# HumanPCR: Probing MLLM Capabilities in Diverse Human-Centric Scenes

Keliang Li[1,3*], Hongze Shen[1,2,3*], Hao Shi[1,2], Ruibing Hou[1], Hong Chang[1,3†]
Jie Huang[1,3], Wen Wang[1,3], Chenghao Jia[1,3], Yiling Wu[2], Dongmei Jiang[2],
Shiguang Shan[1,3], Xilin Chen[1,3]
[1]Institute of Computing Technology, Chinese Academy of Sciences
[2]Peng Cheng Laboratory
[3]University of Chinese Academy of Sciences
`{keliang.li, hongze.shen}@vipl.ict.ac.cn, chonghong@ict.ac.cn`

## Abstract

The aspiration for artificial general intelligence, fueled by the rapid progress of multimodal understanding, demands models to understand humans in diverse and complex scenarios, as humans manifests intelligence and embody the world. We propose **HumanPCR**, an evaluation suite for probing MLLMs' capacity in human-centric visual contexts across three hierarchical levels: **P**erception, **C**omprehension, and **R**easoning (denoted by Human-P, Human-C, and Human-R, respectively). Human-P and Human-C consist of over 6,000 multiple-choice questions evaluating 34 fine-grained tasks covering 9 essential dimensions. Human-R presents a manually curated challenging video reasoning test that requires integrating multiple visual evidence, proactively extracting implicit context beyond question cues, and applying human-like expertise. Each question includes human-annotated Chain-of-Thought (CoT) rationales with key visual evidence to support further research. Extensive evaluations on over 30 state-of-the-art models exhibit significant challenges in human-centric visual understanding, particularly in tasks involving detailed space perception, temporal understanding, and mind modeling. The analysis of Human-R further exposes a critical failure in reasoning: models struggle to proactively gather necessary visual evidence, instead showing a faulty reliance on query-prompted cues, with advanced techniques offering only marginal gains. We hope HumanPCR and our findings will advance the development, evaluation, and human-centric applications of multimodal models.

## 1 Introduction

The rapid advancement of Multimodal Large Language Models (MLLMs) has shown remarkable potential in understanding diverse contexts (Zhang et al., 2024c; Team et al., 2023; Hurst et al., 2024; Bai et al., 2025; Wang et al., 2024). This progress fuels the aspiration toward artificial general intelligence, where a key prerequisite lies in the ability to understand humans in diverse, complex, and dynamic contexts, as human behavior inherently reflects intelligence as well as the complexities of the world (Grauman et al., 2022; Jahangard et al., 2024; Caba Heilbron et al., 2015). In this work, we systematically investigate how well MLLMs understand humans across critical aspects of perception, comprehension, and reasoning in diverse human-centric visual understanding scenarios.

Human-centric visual understanding (Tang et al., 2023; Ci et al., 2023; Li et al., 2025b) remains a fundamental challenge in artificial intelligence. However, current MLLM benchmarks provide a limited assessment of such capabilities. They either isolate narrow tasks, such as action or facial recognition (Qin et al., 2025; Mangalam et al., 2023; Salehi et al., 2024), or, when adopting broader scopes, overlook intricate yet crucial aspects such as gaze and contact, while reporting only coarse-grained scores (Yue et al., 2024; Fu et al., 2024; Xu et al., 2023; Zhou et al., 2024a). Consequently, they lack the necessary **probing power** to rigorously evaluate MLLMs' nuanced capabilities in

---

*Equal contribution.
†Corresponding Author.

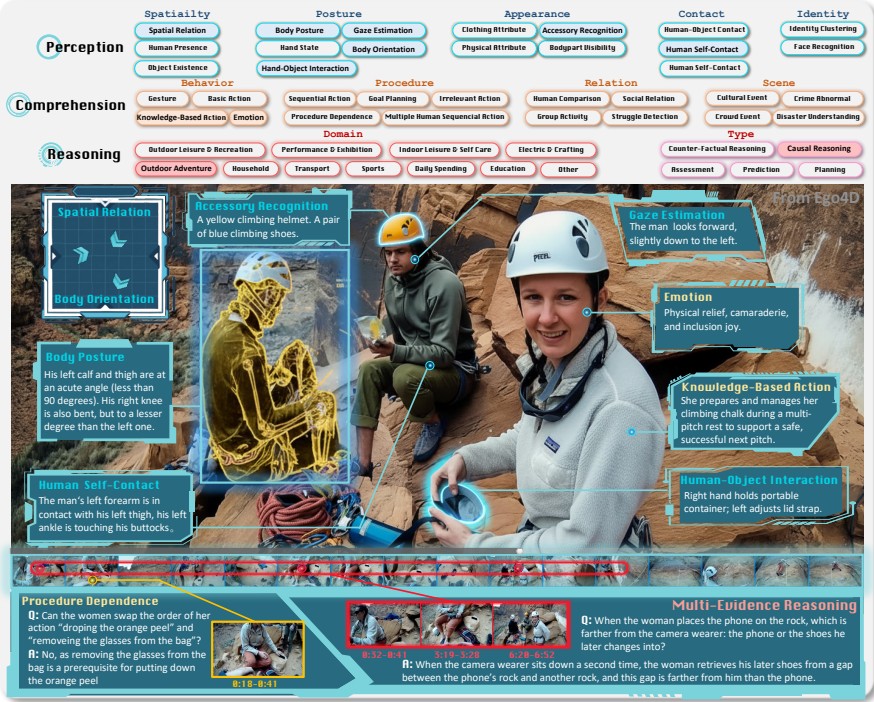

Figure 1: Illustration of HumanPCR, consisting of 34 tasks spanning 9 dimensions across Perception, Comprehension, and Reasoning Levels. It features comprehensive human-centric visual understanding abilities coverage, and proactive visual reasoning based on multiple evidence.

human-centric scenarios, thus providing limited guidance for future research and applications. A more critical gap lies in the assessment of reasoning. Unlike humans, who naturally synthesize multiple visual cues in reasoning, current task-specific or fragmented benchmarks rarely challenge models to perform multi-evidence reasoning (Zhao et al., 2025; Hu et al., 2025; Chen et al., 2024a; Fu et al., 2024; Lu et al., 2023). Although some recent video benchmarks have featured intricate reasoning questions(Cheng et al., 2025; Zhu et al., 2025b; Cai et al., 2025), they often cannot necessitate sophisticated visual evidence demand. As discovered by our analysis in Figure 2, two critical reasoning faculties remain overlooked: 1) the ability of integrating **multiple, disparate visual evidence** to achieve coherent understanding, and 2) the ability of **proactively seeking implicit visual cues**. As a result, reasoning within complex and dynamic human-centric contexts remains a significant, open challenge.

To bridge these critical gaps in evaluation, we introduce **HumanPCR**, a comprehensive evaluation suite designed to meticulously probe the human-centric visual understanding of MLLMs. HumanPCR is structured along a hierarchical taxonomy, perception (**Human-P**), comprehension (**Human-C**), and reasoning (**Human-R**), as illustrated in Figure 1. To enable fine-grained probing, Human-P and Human-C feature a large-scale dataset of over 6,000 image- and video-based QA pairs, assessing 34 tasks that span 9 dimensions from individual attributes to spatio-temporal dynamics. Moreover, Human-R introduces a unique challenge through a manually curated, open-ended video reasoning benchmark. Sourced from 11 diverse human-related domains, it compels models to integrate multiple, disparate visual evidence and proactively seek implicit visual cues beyond what is explicitly prompted. To support further research, each question in Human-R is augmented with expert-annotated Chain-of-Thought (CoT) rationales (Wei et al., 2022) that detail all key visual evidence.

A large suite of open-source and proprietary models is benchmarked on HumanPCR. Our analysis reveals several key findings. First, existing MLLMs face significant challenges in human-centric visual understanding and expose inherent limitations in detailed space perception (Yang et al., 2024b), temporal understanding (Fu et al., 2024), and mind modeling (Rezaei et al., 2025; Jin et al., 2024). Second, Human-R highlights models' struggles with multi-evidence reasoning across diverse human scenes. A substantial proportion of errors arise from their reliance on explicit query-guided cues, failing to proactively seek implicit visual evidence. Third, merely scaling visual contexts offers little gains for Human-R (Team et al., 2023; Hurst et al., 2024; Bai et al., 2025; Wang et al., 2025b; Shen

et al., 2024), emphasizing the need for more precise visual context perception. Conversely, reasoning-enhanced models like o3 (OpenAI, 2025) reduce missed proactive evidence and achieve consistent improvements. We hope that HumanPCR will serve as a crucial tool to accelerate the development of more capable MLLMs and facilitate their adaptation to diverse human-related applications.

## 2 RELATED WORK

**Multimodal Large Language Models (MLLMs).** Evolving from LLMs, MLLMs now process diverse modalities including image sequences, video, and audio (Liu et al., 2023; Li et al., 2024a; Lin et al., 2023; Borsos et al., 2023). Recent models can handle dynamic resolutions and long contexts (Bai et al., 2025; Zhu et al., 2025a). To manage long videos within limited context windows, common strategies include efficient content extraction via frame selection (Tang et al., 2025; Huang et al., 2025; Ye et al., 2025) or token pruning (Wang et al., 2025b; Zhang et al., 2025; Shen et al., 2024). Concurrently, the rise of models with powerful intrinsic reasoning capabilities (Guo et al., 2025; Jaech et al., 2024; Comanici et al., 2025; OpenAI, 2025) has inspired efforts to leverage these cognitive strengths for more advanced visual understanding (Feng et al., 2025; Xu et al., 2024; Wang et al., 2025a; Li et al., 2025a).

**Benchmarks for MLLM Evaluation.** Evaluating MLLMs, particularly in human-centric contexts, is a significant research focus. Many benchmarks target specific human-related tasks like motion analysis (Hong et al., 2025; Feng et al., 2023), face recognition (Qin et al., 2025; Pham et al., 2024), or domain-specific actions (Salehi et al., 2024; Cui et al., 2023; Plizzari et al., 2025), while they have concluded that MLLMs still have substantial limitations in nuanced human understanding (Rezaei et al., 2025; Mangalam et al., 2023; Zhou et al., 2024b; Li et al., 2024c). Broader, general benchmarks also include human-centric tasks but often lack the structured taxonomies needed for systematic, fine-grained ability diagnosis (Xu et al., 2023; Fu et al., 2024; Li et al., 2023). On the reasoning front, benchmarks have grown in complexity, moving from image-based exams (Yue et al., 2024; Chen et al., 2024a) to challenging video-based scenarios (Song et al., 2025; Cai et al., 2025; Cheng et al., 2025; Zhu et al., 2025b). However, they often focus on knowledge-based reasoning from instructional videos (Hu et al., 2025) or repurpose existing QA pairs (Han et al., 2024; Qi et al., 2025), which may not adequately test the synthesis of multiple, disparate visual cues. Our HumanPCR, features a hierarchical taxonomy for detailed ability diagnosis and assessing the critical skills of integrating multiple pieces of visual evidence and proactively seeking implicit information in complex, human-centric videos.

## 3 THE HUMANPCR BENCHMARK

### 3.1 TASK ARCHITECTURE AND DATA CONSTRUCTION

We introduce the HumanPCR benchmark to evaluate how MLLMs understand humans in real-world scenarios. The design of HumanPCR is motivated by **the need for probing fine-grained model capability essential to downstream applications** in human-centric visual understanding. To this end, a critical principle is to construct sufficient number of fine-grained tasks with comprehensive coverage and low redundancy. And each task should be supported by data sources as rich and varied as possible. To do so, we survey a wide range of human-centric perception and understanding works to first define tasks, then match them with rich and varied datasets. This iterative approach, grounded in diverse data from daily to professional scenes, mitigates single-source bias and ensures broad capability coverage (Figure 3). Task definitions and sources are provided in Supplementary Materials. The taxonomy is briefly introduced as follows:

**Level 1: Perception** evaluates visual recognition across 5 dimensions and 17 tasks: (1) **Spatiality**: perceiving existence of people, objects, and their spatial relations; (2) **Posture**: recognizing physical position and orientation of body parts, hands, and gaze; (3) **Appearance**: identifying human appearances, including inherent attributes and attirement; (4) **Contact**: recognizing detailed interaction regions between people and objects, or themselves; (5) **Identity**: recognizing people's identity.

**Level 2: Comprehension** assesses visual concepts comprehension, from 4 dimensions and 17 tasks, based on commonsense or domain-specific cues: (1) **Behavior**: understanding human actions and

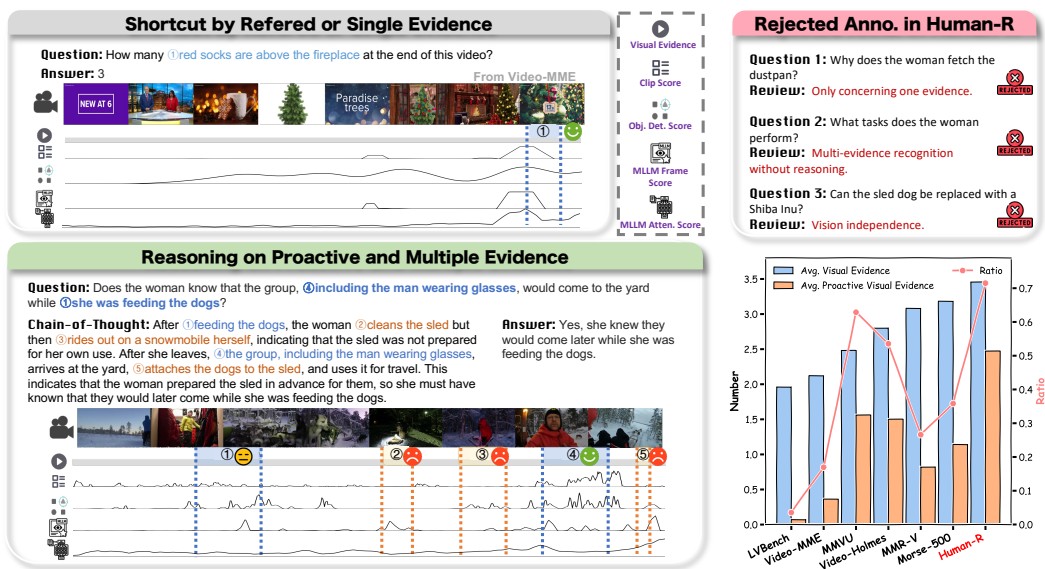

Figure 2: Illustration of multiple and proactive evidence reasoning in Human-R. (Left) Examples from Video-MME and Human-R. Clips of proactive and referred *visual evidence* are highlighted. Context matching by heuristics could resolve sparse or referred evidence and bypass comprehensive reasoning, so it is crucial to assess reasoning over multiple and proactive evidence. (Right) By rigorously filtering out annotations that fall below the required reasoning complexity, Human-R can precisely diagnose a model's capabilities in video reasoning with multiple and proactive evidence, an area where existing benchmarks fall short.

body movements, such as gestures and emotions; (2) **Procedure**: thoroughly understanding long-term activities, including underlying intentions and dependence among action sequences; (3) **Relation**: analyzing relations, roles and differences among individuals; (4) **Scene**: interpreting group dynamics or human activities within broader contexts.

**Level 3: Reasoning** examines whether models can integrate continuous, tightly coupled human dynamics within complex scenes for reasoning. We contend that the evaluation should satisfy three criteria: (1) **Visual Complexity**: questions should require sufficient *visual evidence*[1] , and exclude redundant content, going beyond simple concept retrieval; (2) **Reasoning Necessity and Diversity**: questions should engage diverse reasoning chains rather than be limited to a few reasoning patterns; (3) **Proactivity**: questions should demand proactive extraction of visual evidence over the abundant contexts[2], rather than relying solely on the referred evidence in the question.

For the Reasoning level (Human-R), we found that videos from public academic datasets are not diverse enough and usually short. Therefore, we defined 11 domains ranging from daily life to professional scenarios, as detailed in Figure 1. Web videos were collected as a supplementary data source to populate these domains. They were pre-filtered via domain-relevant tags and then manually reviewed before annotation to guarantee both content richness and safety.

## 3.2 QA ANNOTATION PROTOCOL

The prompt and other details of annotation process are provided in Appendix B.

**Human-P** and **Human-C.** Benefiting from the data collection, we leverage annotations from existing datasets to efficiently scale up QA pairs without compromising data quality or diversity. Templates- and LLM-based generation are used to create *Multi-Choice questions* and options based on dataset annotations, as illustrated in the lower-left panel of Figure 3. Moreover, for under-explored tasks in

---

[1]In this work, *visual evidence* refers to information conveyed in images or videos, such as instance attributes or visual relationship, which serve as information units or propositions in reasoning.

[2]"Proactive visual evidence" refers to visual information that is not, or only partially, cued by the question, in contrast to "referred visual evidence" which is explicitly indicated by the question.

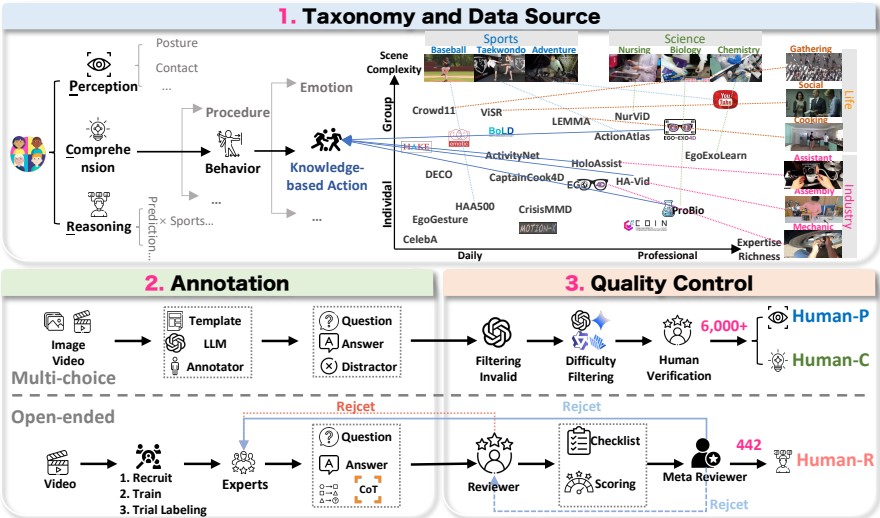

Figure 3: A comprehensive pipeline for HumanPCR Construction. The process includes: (1) building a hierarchical task taxonomy through surveys and task-driven data collection to ensure diversity, (2) recruiting annotators and conducting task-specific annotations, and (3) hybrid automated–manual verification with iterative refinement.

existing datasets, we manually generate QA pairs and complementary annotations with the assistance of domain-specific expert annotators.

**Human-R.** Domain experts were recruited to annotate *open-ended questions* that encompass 5 distinct types of reasoning: Causal Reasoning, Prediction, Counter-Factual Reasoning, Assessment, and Planning. To satisfy our core reasoning criteria above, answering the questions requires integrating **multiple pieces of visual evidence**, engaging diverse reasoning chains, and extracting at least one **proactive visual evidence** from the video. In this process, the detailed *reasoning steps* and the necessary *visual evidence* were also annotated. As illustrated in Figure 2, this design reliably tests a model's ability for holistic multimodal reasoning based on accurate evidence perception. This stands in contrast to tests relying on sparse evidence or fully specified refernces, which often lead to shortcut solutions instead of genuine video understanding and sufficient reasoning.

### 3.3 QUALITY CONTROL AND VERIFICATION

For **Human-P** and **Human-C**, QA pairs are first filtered by LLMs to eliminate those solvable without visual input, followed by human verification conducted by trained annotators. Each annotation is carefully reviewed for linguistic quality, answer accuracy, distractor plausibility, and, most importantly, its reliance on visual context. For **Human-R**, reviewers firstly fill out a detailed checklist that validates annotations' objectivity, factual accuracy, non-redundancy, and complexity; they then assign a quantitative score and deliver targeted feedback to the annotator for a chance to modify. Meta-reviewers **further assess question complexity**, ensuring that every question (1) requires integrating multiple visual evidence, which cannot be fully determined from the question alone, and (2) relies on at least one essential proactive visual evidence. The interaction flow among the annotators, reviewers, and meta-reviewers is illustrated in the lower-right panel of Figure 3.

This pipeline yields over 6,000 Human-P&C multiple-choice questions, and 442 Human-R open-ended questions with a final acceptance rate 20%. Table 2 and Figure 4 summarize HumanPCR 's scale and modality diversity, while Figures 2 and 5 show that Human-R demands strong visual evidence across all video lengths, well above prior datasets, reflecting rigorous quality control.

### 3.4 COMPARISON WITH EXISTING BENCHMARKS

As summarized in Table 1, HumanPCR fills critical gaps across two key domains:
(1) **Human-centric Benchmarks:** Existing works are often either too narrow, limited to scopes like action (Hong et al., 2025; Salehi et al., 2024), or too broad, targeting general visual understanding

Table 1: Comparison of HumanPCR and existing benchmarks with respect to: the number of ability dimensions (**#Dim.**) and tasks (**#Tasks**), covered modalities (**Mod.**), human-centric orientation (**HC**), taxonomy for probing and diagnostic analysis (**Probing**), average video duration (**V.Len.**), method of annotation (**Anno.**, M/A means human-annotated/ automatically generated), open-ended questions (**OE**), sourced from a broad range of open domains (**OD**), availability of rationales (**CoT**), requirement for multiple visual evidence (**MVE**) and proactive visual reasoning (**Proactive**).

| Benchmarks | #Dim. | #Tasks | Mod. | HC | Probing | OD |
|---|---|---|---|---|---|---|
| MotionBench | - | 1 | V | ✓ | ✗ | ✗ |
| Face-Human-Bench | 4 | 18 | I | ✓ | ✗ | ✗ |
| ActionAtlas | - | 1 | V | ✓ | ✗ | ✗ |
| MME | 4 | 14 | I | ✗ | ✗ | ✓ |
| MVBench | 9 | 20 | V | ✗ | ✓ | ✓ |
| HumanVbench | 4 | 16 | V | ✓ | ✗ | ✗ |
| **Human-P&C** | **9** | **34** | **I+V** | ✓ | ✓ | ✓ |

| Benchmarks | V.Len. | Anno. | OE | OD | HC | CoT | MVE | Proactive |
|---|---|---|---|---|---|---|---|---|
| EgoSchema | 180 | M | ✗ | ✗ | ✓ | ✗ | ✓ | ✗ |
| VideoMME | 1017 | M | ✗ | ✓ | ✗ | ✗ | ✗ | ✗ |
| CG-Bench | 1624 | M | ✓ | ✓ | ✗ | ✗ | ✗ | ✗ |
| LVbench | 4101 | M | ✗ | ✓ | ✗ | ✗ | ✗ | ✗ |
| MMVU | 51 | M | ✗ | ✗ | ✗ | ✓ | ✗ | ✗ |
| Video-Holmes | 60~300 | A | ✗ | ✗ | ✗ | ✓ | ✓ | ✗ |
| MMR-V | 277 | A | ✗ | ✓ | ✗ | ✗ | ✓ | ✗ |
| **Human-R** | **469.3** | **M** | ✓ | ✓ | ✓ | ✓ | ✓ | ✓ |

Table 2: Key statistics of HumanPCR.

| | Statistics | Value(Avg./Max) |
|---|---|---|
| Human-P&C | # Multi-Choice | 6176 |
| | # Options | 4.9 / 5 |
| | # Images | 1.1 / 6 |
| | Video Duration | 35.4 / 584.1 |
| Human-R | # Open-Ended | 442 |
| | Question Length | 19.2 / 79 |
| | CoT Length | 86.2 / 183 |
| | Answer Length | 18.6 / 64 |
| | Video Duration | 469.3 / 5225.0 |

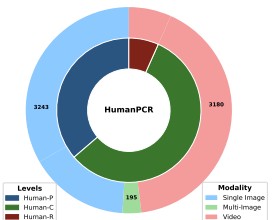

Figure 4: Modality distribution in HumanPCR.

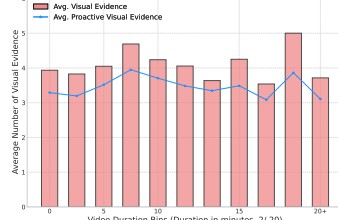

Figure 5: The distribution of the visual evidence number.

without alignment to human-centric tasks (Xu et al., 2023; Li et al., 2024d). Even dedicated human-centric benchmarks (Qin et al., 2025; Zhou et al., 2024b) like HumanVBench are restricted by a few dimensions and a single modality. HumanPCR provides a comprehensive and fine-grained taxonomy for probing human-centric capabilities across a diverse range of tasks and modalities.

(2) **Video Reasoning Benchmarks:** General video benchmarks typically assess shallow comprehension with limited reasoning diversity and depth (Fu et al., 2024; Zhou et al., 2024a; Wu et al., 2024). Early works either struggle with question quality due to semi-automatic annotation (Qi et al., 2025; Han et al., 2024) or are confined to narrow source or task (Jin et al., 2024; Song et al., 2025). While concurrent VideoHolmes (Cheng et al., 2025) and MMRV (Zhu et al., 2025b) require multi-frame reasoning, their task-specific designs limit the diversity of evidence and reasoning paths. Most importantly, their multiple-choice or certain question references often implicitly reveal the required visual evidence, failing to evaluate the crucial ability of proactive evidence seeking. Human-R is thus distinguished by its multi-domain, open-ended design that uniquely demands both the integration of multiple visual cues and the proactive search for implicit evidence.

# 4 EXPERIMENTS

## 4.1 EVALUATION SETTINGS

**(1)Models.** We benchmark a diverse set of models, including 9 proprietary (e.g., Gemini-2.5-Flash, o4-mini) and 30 open-source MLLMs (e.g., Qwen-VL, InternVL) on HumanPCR. On Human-R, we add models specialized for video understanding and thinking. Human performance is also provided for comparison (Appendix C.4). **(2)Configuration.** For evaluation, we employ **Direct Answer** prompts for multi-choice questions and **CoT** for open-ended ones. Video inputs are processed by sampling 32 frames for multi-choice tasks and the maximum allowable frames for open-ended tasks. Further details on model configurations and prompts are in Appendix C.1 and C.2. **(3)Metrics.** Accuracy for multi-choice questions is determined by matching responses to the correct option. The average accuracy for each task, and the macro-average accuracy of tasks for each dimension or level is reported. For open-ended questions, we use a proprietary model, o3-mini, as a judge, which demonstrates high agreement with human evaluations (Sec 4.4).

Table 3: Results of Proprietary and Open-Source Models on HumanPCR. Macro-average accuracy of tasks within each level and dimension is reported. "†" indicates that the reported performance is based only on a subset of the HumanPCR dataset. Full results is in Appendix F.

| Models | Multi-Choice | | | | | | | | | | | | Open |
|---|---|---|---|---|---|---|---|---|---|---|---|---|---|
| | Human-P | | | | | | Human-C | | | | | | Human-R |
| | Spa. | Pos. | App. | Con. | Ide. | avg. | Beh. | Pro. | Rel. | Sce. | avg. | Acc. | Acc. |
| Random | 20.00 | 20.00 | 20.00 | 20.00 | 35.00 | 23.00 | 21.00 | 20.00 | 20.00 | 20.00 | 20.25 | 21.78 | 0.00 |
| Human† | 89.98 | 81.44 | 87.02 | 87.28 | 96.43 | 88.43 | 69.24 | 81.42 | 78.95 | 65.83 | 73.86 | 81.95 | 73.17 |
| **Open-source Models** | | | | | | | | | | | | | |
| Aria (Li et al., 2024b) | 79.12 | 39.72 | 61.32 | 36.58 | 76.53 | 55.53 | 48.85 | 41.31 | 50.08 | 55.30 | 48.44 | 51.98 | 28.96 |
| LongVILA-256 (Chen et al., 2024b) | 75.96 | 31.84 | 63.28 | 38.87 | 41.63 | 49.41 | 44.97 | 31.91 | 51.12 | 53.20 | 44.51 | 46.96 | 21.49 |
| NVILA-8B (Liu et al., 2024a) | 76.39 | 35.40 | 64.53 | 42.53 | 47.95 | 52.22 | 44.10 | 28.59 | 50.69 | 53.18 | 43.23 | 47.72 | 22.17 |
| MiniCPM-V-2.6 (Yao et al., 2024) | 74.07 | 36.90 | 57.77 | 35.56 | 70.47 | 52.08 | 47.39 | 28.66 | 50.46 | 54.67 | 44.32 | 48.20 | 16.74 |
| MiniCPM-o-2.6 (OpenBMB, 2024) | 80.08 | 43.80 | 63.58 | 36.95 | 71.26 | 56.88 | 47.83 | 31.43 | 53.98 | 55.38 | 46.23 | 51.56 | 21.04 |
| LLaVA-Video-7B (Zhang et al., 2024c) | 74.73 | 37.13 | 58.03 | 32.62 | 63.50 | 50.99 | 46.18 | 36.26 | 49.54 | 51.77 | 45.37 | 48.18 | 27.60 |
| LLaVA-Video-72B (Zhang et al., 2024c) | 76.64 | 42.97 | 71.68 | 56.54 | 63.63 | 60.49 | 51.74 | 43.73 | 55.41 | 60.92 | 52.41 | 56.45 | 28.28 |
| LLaVA-OneVision-7B (Li et al., 2024a) | 77.95 | 36.33 | 60.03 | 34.13 | 54.63 | 51.02 | 47.67 | 35.56 | 52.40 | 51.06 | 46.02 | 48.52 | 22.85 |
| LLaVA-OneVision-72B (Li et al., 2024a) | 82.57 | 44.77 | 70.40 | 57.79 | 71.95 | 62.96 | 51.53 | 43.19 | 58.17 | 61.52 | 52.99 | 57.98 | 27.60 |
| Oryx-1.5-7B (Liu et al., 2024b) | 74.62 | 36.48 | 62.36 | 42.49 | 39.18 | 50.68 | 44.21 | 34.37 | 51.98 | 49.21 | 44.32 | 47.50 | 22.17 |
| Oryx-1.5-32B (Liu et al., 2024b) | 82.08 | 40.86 | 64.88 | 47.90 | 45.00 | 55.52 | 47.61 | 44.61 | 54.53 | 57.51 | 50.68 | 53.10 | 28.51 |
| Qwen2.5-VL-7B (Bai et al., 2025) | 78.66 | 42.20 | 62.33 | 30.68 | 41.32 | 51.23 | 49.73 | 32.66 | 51.89 | 55.83 | 46.65 | 48.94 | 26.24 |
| Qwen2.5-VL-72B (Bai et al., 2025) | 82.11 | 50.00 | 68.70 | 43.76 | 41.32 | 57.94 | 55.77 | 46.49 | 53.23 | 64.43 | 54.48 | 56.21 | 34.39 |
| InternVL2.5-8B (Chen et al., 2024c) | 79.43 | 41.68 | 60.37 | 36.69 | 75.45 | 55.83 | 48.53 | 39.13 | 52.24 | 56.47 | 48.51 | 52.17 | 23.53 |
| InternVL2.5-38B (Chen et al., 2024c) | 84.34 | 45.14 | 68.70 | 50.02 | 84.29 | 63.07 | 53.89 | 55.79 | 54.44 | 63.17 | 56.76 | 59.92 | 35.97 |
| InternVL2.5-78B (Chen et al., 2024c) | 84.80 | 44.68 | 69.31 | 50.87 | 81.24 | 62.95 | 57.54 | 53.93 | 57.24 | 65.20 | 58.21 | 60.58 | 33.94 |
| InternVL3-8B (Zhu et al., 2025a) | 81.64 | 40.54 | 67.20 | 38.22 | 72.84 | 57.46 | 51.39 | 40.86 | 55.00 | 59.15 | 50.97 | 54.21 | 31.45 |
| InternVL3-38B (Chen et al., 2024d) | 84.73 | 49.55 | 69.16 | 58.08 | 85.89 | 66.15 | 57.86 | 55.38 | 59.01 | 66.02 | 59.32 | 62.74 | 35.75 |
| InternVL3-78B (Zhu et al., 2025a) | 86.54 | 46.46 | 73.39 | 50.82 | 86.42 | 65.34 | 57.96 | 54.31 | 59.78 | 70.21 | 60.20 | 62.77 | 37.56 |
| **Proprietary Models** | | | | | | | | | | | | | |
| Doubao-1.5-vision-pro (ByteDance, 2024) | 72.50 | 36.96 | 67.19 | 37.61 | 78.29 | 55.32 | 45.96 | 45.20 | 46.00 | 53.67 | 47.56 | 51.44 | 32.81 |
| Grok-2-Vision (xAI, 2024) | 57.83 | 30.97 | 57.51 | 41.14 | 50.21 | 46.01 | 46.51 | 42.42 | 42.27 | 56.52 | 46.67 | 46.34 | 36.20 |
| Claude-3.5-Sonnet-v2 (Anthropic, 2024) | 67.99 | 39.84 | 59.35 | 44.68 | 66.26 | 53.36 | 50.39 | 46.88 | 49.08 | 59.19 | 51.12 | 52.24 | 39.59 |
| Gemini-1.5-Flash (Team et al., 2024) | 54.99 | 38.34 | 53.78 | 32.82 | 54.45 | 45.83 | 47.63 | 41.92 | 44.64 | 51.81 | 46.23 | 46.03 | 35.97 |
| Gemini-1.5-Pro (Team et al., 2024) | 66.80 | 45.62 | 56.04 | 39.34 | 69.03 | 53.45 | 51.67 | 44.84 | 50.16 | 61.81 | 51.69 | 52.57 | 40.05 |
| Gemini-2.0-Flash (Google DeepMind, 2024) | 76.42 | 47.46 | 63.75 | 52.32 | 73.08 | 60.28 | 53.09 | 42.27 | 54.37 | 60.76 | 52.01 | 56.14 | 38.01 |
| Gemini-2.5-Flash (Comanici et al., 2025) | 82.01 | 48.10 | 69.41 | 49.37 | 93.50 | 64.66 | 54.05 | 49.19 | 57.27 | 62.56 | 55.38 | 60.02 | 43.44 |
| GPT-4o (Hurst et al., 2024) | 70.14 | 40.56 | 55.46 | 33.60 | 35.11 | 47.41 | 52.01 | 38.93 | 48.85 | 60.14 | 49.33 | 48.37 | 41.40 |
| o4-mini (OpenAI, 2025) | 80.69 | 53.13 | 71.86 | 41.09 | 85.89 | 64.13 | 61.10 | 54.43 | 61.68 | 65.97 | 60.42 | 62.28 | 53.39 |

## 4.2 MAIN RESULTS

Table 3 presents the main results of HumanPCR. Our principal findings are summarized as follows:
**Current MLLMs are far from reliable human-centric understanding.** A significant gap exists across all levels, with humans outperforming the best MLLMs by over 15%. Leading models like InternVL3-78B achieve average accuracies of only 63.66% on Human-P&C, lagging behind the human baseline of 81.95%. While models show promise in basic tasks like *Spatiality*, the deficits in fine-grained perception (*Posture*, *Contact*) and high-level understanding (*Procedure*, *Relation*) underscore the need for the detailed, probing evaluation that HumanPCR provides.

**Open-source models rival in perception and comprehension, lag in reasoning.** On Human-P and Human-C, open-source models match proprietary ones. Notably, InternVL3-78B surpasses the top proprietary model, o4-mini. However, they underperform in reasoning—most remain below 30% accuracy on Human-R, whereas all proprietary models exceed this value. Models like o4-mini and Gemini-2.5-Flash illustrate the advantages of proprietary designs for reasoning. While open-source models match proprietary ones at perception and comprehension, they struggle on reasoning tasks involving complex visual evidence.

**Understanding human-centric scenes reflects general capabilities.** The consistent poor performance on dimension *Posture*, *Contact*, *Behavior*, and *Procedure* points to fundamental limitations that transcend human-centric scenarios. These results expose core deficiencies in fine-grained spatial perception (especially with occlusion) and in the temporal understanding necessary to complex, long-term activities. Therefore, the challenges presented in HumanPCR do not merely identify gaps in human-centric understanding but also reflect general, critical shortcomings in current MLLMs.

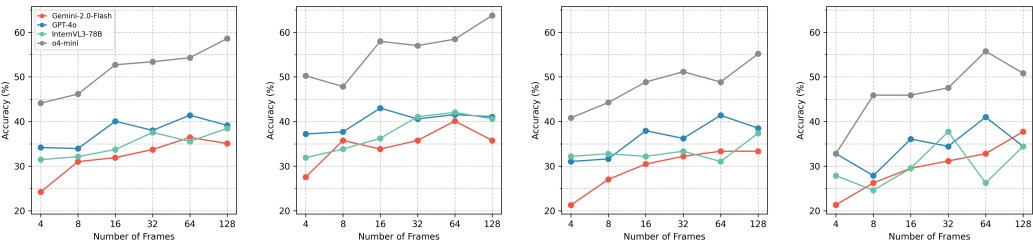

| (a) Overall Acc. | (b) Visual Evidence: 2-3 | (c) Visual Evidence: 4-5 | (d) Visual Evidence: 6+ |

Figure 6: Effect of frame sampling on Human-R. (a) Overall accuracy. (b–d) Accuracy grouped by the number of visual evidence required in the question: (b) 2–3 evidence, (c) 4–5 evidence, (d) 6 or more evidence. Increasing frames shows varying impact depending on visual complexity.

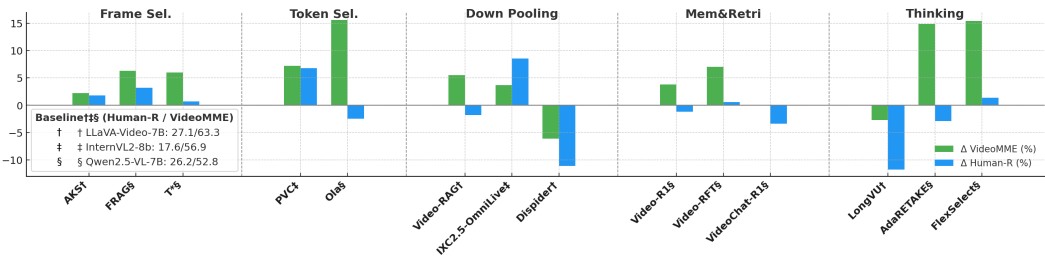

Figure 7: Accurary gain over video understanding baseline methods on Human-R and Video-MME. See the full configuration of them in Appendix C.1

## 4.3 DELVING INTO VISUAL REASONING ON MULTIPLE EVIDENCE

On Human-R, we analyze the characteristics and limitations of MLLMs' reasoning ability.

**Effects of frame number.** As shown in Figure 6, merely increasing the input frame count yields negligible accuracy gains for most models. Notably, only the reasoning-oriented o4-mini improves consistently, implying that stronger reasoning is needed to make full use of broader visual context. This trend is amplified as the reasoning challenge intensifies: when a problem requiring integrating more pieces of visual evidence, overall accuracy drops. In these more complex scenarios, the marginal benefit of adding frames diminishes, even becoming detrimental in cases requiring six or more evidence items. This suggests a larger visual context can introduce distractors, complicating the evidence extraction and integration process when the core reasoning task becomes more demanding.

> **Takeaway:** *More frames may mitigate perceptual gaps, but cannot substitute for the core reasoning ability required to effectively utilize broader visual contexts.*

**Does advanced video understanding configuration help?** We investigate advanced configurations to determine if they could address the challenges in Human-R. (1) **Visual context extraction**. As shown in Figure 7, techniques that are effective on standard benchmarks like Video-MME (Fu et al., 2024), such as token selection and memory-based retrieval, see a significant performance drop on Human-R. This performance gap verifies that integrating multiple visual cues required by Human-R presents a unique challenge that cannot be solved by simple query-guided or heuristic matching methods. (2) **Test-time computation**. Best-of-N (BoN) sampling consistently boosts performance by over 5%, with gains scaling with better reward models. In contrast, Self-Refine offers only marginal benefits. The strong performance of specialized reasoning models like Video-R1 and the proprietary o3 (Figure 7, Table 4) underscores that enhancing the core reasoning process itself is critical.

> **Takeaway:** *Complex reasoning demands finer context management than general understanding; simple, heuristic extraction fail to generalize to problems requiring multi-faceted evidence.*

**Error analysis.** Manual analysis of errors within a subset of 200 questions for top models on Human-R (shown in Figure 8a) reveals the predominant failure is visual-evidence extraction, specifically in identifying proactive visual evidence (in Figure 8b). This suggests models often treat the question as a retrieval shortcut, bypassing necessary complex reasoning. Our findings support that while scaling the frames significantly reduces errors of Gemini-2.0-flash in missing referred evidence, it provides

Table 4: Results for test-time scaling strategies on Human-R. BoN uses $2^M$ candidates and Self-Refine performs $M$ iterations. Results of more models are in Table12.

| Model | Method | Reward Model | Direct | M=0 (CoT) | M=1 | M=2 | M=3 |
|---|---|---|---|---|---|---|---|
| GPT-4o | BoN | GPT-4o (Self) | 32.81 | 41.40 | 43.21 | 45.70 | 45.93 |
| | BoN | Gemini-2.5-Flash | | | 44.57 | 45.02 | **46.38** |
| | BoN | o4-mini | | | 44.11 | 46.38 | 45.02 |
| | Self-Refine | - | | | 39.59 | 39.82 | 40.95 |
| Gemini-2.5-Pro | Thinking | - | - | 51.13 | | - | |
| Claude-3.7-sonnet | Thinking | - | - | 40.50 | | - | |
| o3 | Thinking | - | - | **59.28** | | - | |

Table 5: Benchmark evaluation across input modalities on GPT-4o. 'Image' uses the central frame; 'Video' samples 32 uniform frames.

| Benchmark | Text (ratio↓) | Image (ratio↓) | Video |
|---|---|---|---|
| Video-MME | 44.00 (61.1%) | 54.14 (75.0%) | 71.85 |
| LongVideoBench | 40.38 (69.0%) | 41.70 (71.3%) | 58.50 |
| **Human-R** | 2.94 (**7.3%**) | 11.08 (**26.8%**) | 41.40 |

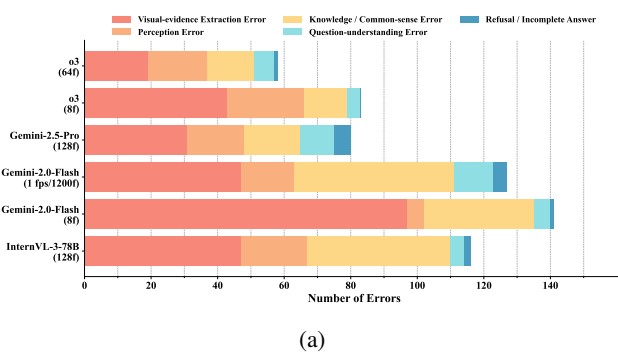

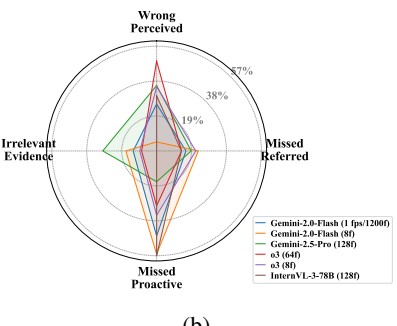

(a)                                           (b)

Figure 8: Distribution of error types on Human-R across top models ("f": the number of sampled frames). (a) Counts of 5 major error types. Most errors are visual-related. (b) Fine-grained breakdown of visual-related errors. Proactive evidence is more frequently missed than referred evidence.

minimal benefit for finding proactive evidence. Consequently, models with superior reasoning, such as Gemini-2.5-Pro (Comanici et al., 2025) and Claude-3.7-sonnet (Anthropic, 2025), outperform high-frame-count models like Gemini-2.0-Flash, even with fewer frames. Furthermore, models exhibit distinct error tendencies: For example, while Gemini-2.5-Pro reduces proactive evidence omissions compared to o3, it tends to introduce more irrelevant information.

> **Takeaway:** *Proactive evidence extraction is a major practical challenge on Human-R, and difficulties in selecting implicit visual cues can limit performance, suggesting that evaluation setups should discourage purely query-driven shortcuts.*

**Interventions on evidence extraction difficulty.** To further probe the role of proactive evidence, we conduct an intervention study on Human-R, progressively lowering the difficulty of visual evidence extraction while keeping the question and answer fixed. We enrich prompts with three levels of guidance: (1) **Relation awareness** (Level 1), giving generic relation-type hints (e.g., "check surrounding context"); (2) **Logic awareness** (Level 2), additionally highlighting which referred cues are logically linked to potential proactive evidence; and (3) **Proactive guidance** (Level 3), adding vague descriptions that loosely point to the proactive evidence without revealing the reasoning steps or answer. As shown in Table 6, Level 1/2 hints yield only modest gains, whereas Level 3 consistently improves accuracy by about 10–13 points across models. This indicates that directly easing proactive evidence extraction has a much larger impact than generic relational or logical hints, providing intervention-based support for our insight that proactive evidence extraction is a prominent practical weakness of current models on complex video reasoning.

> **Takeaway:** *On Human-R, vague guidance that directly targets proactive evidence yields the largest gains, reinforcing proactive evidence extraction as a prominent practical weakness in complex video reasoning.*

## 4.4 FURTHER ANALYSIS

**Mixed CoT effects and diagnostic value of Human-P/C.** Figure 9a shows that CoT has mixed effects on Human-P/C: proprietary models typically improve, whereas many open-source models see limited or even negative gains, and the trend already varies across high-level dimensions, with visually complex and understanding-related questions benefiting more than simple attribute-perception tasks. Figure 9b further reveals substantial spread in both accuracy and Δ (cot–direct) across neighboring

Table 6: Intervention on evidence extraction for Human-R with various evidence hints. Level 1/2/3 provide generic relational, logical hints, and vague descriptions of proactive evidence, respectively.

| Model | Orig. | Level 1 | Level 2 | Level 3 |
|---|---|---|---|---|
| o4-mini | 53.39 | 52.26 (-1.13) | 54.07 (+0.68) | 63.35 (+9.96) |
| GPT-4o | 41.40 | 41.62 (+0.22) | 44.11 (+2.71) | 52.35 (+10.95) |
| Gemini-2.5-Flash | 43.44 | 41.40 (-2.04) | 45.93 (+2.49) | 53.40 (+9.96) |
| Qwen-2.5-VL-72B | 34.39 | 34.61 (+0.22) | 38.46 (+4.07) | 47.74 (+13.35) |

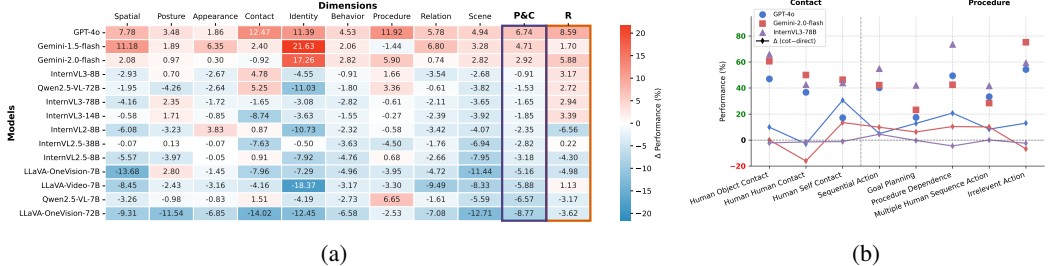

(a)                                                                (b)

Figure 9: Ablation study examining the impact of CoT. (a) Relative improvements of CoT of models on all dimensions and levels. (b) Detailed performance across tasks in the *Contact* and *Procedure* dimensions. "$\Delta$ (cot-direct)" represents the accuracy difference between direct answers and CoT-prompted responses.

subtasks within the same dimension, with some tasks consistently benefiting from CoT while others degrade and model rankings change accordingly. These heterogeneous patterns indicate that Human-P/C subtasks are non-redundant and provide diagnostic resolution that would be lost under a single coarse "action" or "relation" score.

**Human-R Quality Check** To validate the quality of Human-R, we examine potential single-frame and textual biases and compare them with other impactful benchmarks. As shown in Table 5, Human-R exhibits **minimal bias**, unlike datasets such as Video-MME where strong results can be achieved using only text or a single frame, revealing redundancy and bias in evaluating full video reasoning. This confirms that Human-R tasks require genuine temporal and multi-evidence reasoning, demonstrating the effectiveness of our rigorous curation and expert review.

**Reliability of LLM-based Judges.** To verify evaluation robustness, we compared multiple LLM judges against 4000 human annotations, as in Figure 10. All judges produced highly consistent model rankings, with strong Pearson correlations to human accuracy (higher than 0.64) and strong alignment on pairwise win-fail preferences (around 0.85). Additional analyses(Appendix C.3) show that annotated CoT could improve agreement with humans while no self-preference of LLM judge is observed.

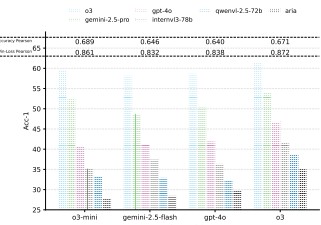

Figure 10: Model performance under multiple judge models and their agreement to Human judge.

## 5 CONCLUSION

We present HumanPCR, a comprehensive benchmark for evaluating how well MLLMs understand humans across diverse scenarios. First, it probes MLLMs' nuanced visual understanding in human-centric scenarios through a systematic, fine-grained taxonomy. Second, it introduces a paradigm for video reasoning that integrates disparate visual evidence and proactively seeks implicit visual cues. Consequently, HumanPCR reveals persistent shortcomings and yields diagnostic insights. Specifically, MLLMs not only face challenges detailed space perception, temporal understanding, and mind modeling, but also often fail to proactively extract visual evidence in reasoning. Limitations of HumanPCR include reliance on academic datasets; future work will extend to professional domains and efficient annotation methods.

ACKNOWLEDGMENTS

This work was partially supported by the National Natural Science Foundation of China (Nos. 62461160331, 62306301, and 62576334). We also thank Dingcun Chen, Yansong Li, Cunku Liang, Ailin Zhao, Weiming Liu, Jun Zhang, Yutong Shen, Yuhan Wang and Han Liu for their valuable suggestions and feedback on the annotation protocol of our dataset.

## ETHICS STATEMENT

We have prioritized ethical considerations throughout the creation and planned release of HumanPCR. 1. **Data Sourcing and Privacy**: Our benchmark is built upon previously published, public datasets and supplemented with publicly available internet videos. To respect copyright and the privacy of content creators, we will only release metadata (e.g., public video IDs and timestamps) for internet-sourced videos, not the raw video files themselves. Our full data hosting and usage policy is detailed in our Data Use Agreement (DUA) in Figure 25. 2. **Annotation and Distribution Safeguards**: During the annotation process, all annotators and reviewers were instructed to filter out questions that require time-sensitive, private, or personally identifiable information for their resolution. The HumanPCR benchmark is intended strictly for non-commercial, academic research purposes. The DUA explicitly prohibits any use of the data for biometric identification, tracking, surveillance, or the development of related applications. 3. **Compliance and Responsibility**: For any source data that we are permitted to host, we require end-users to agree to the original dataset's license terms. We are committed to being responsible stewards of this data and will promptly respond to any inquiries or takedown requests from copyright holders.

## REPRODUCIBILITY STATEMENT

We are committed to ensuring the full reproducibility of our work. 1. **Code and Prompts**: All prompts used for model evaluation are detailed in Appendix C.2. The evaluation code has been included in the supplementary materials. Configurations for all evaluated models are detailed in Appendix C. 2. **Data and Annotations**: All annotations created for HumanPCR will be released to public in accordance with our DUA upon acceptance. For the small subset of source data we cannot host directly (due to original license or copyright), full metadata and data processing scripts would be provided to reconstruct the dataset. Our data release policy has been detailed in Appendix E. A portion of the data is available in the supplementary material to demonstrate its structure.

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

## Summary of Appendix

The appendix is organized as follows:

- Details of the Taxonomy (Appendix A);
- Annotation Setup (Appendix B);
- Evaluation Setup (Appendix C);
- Additional Data Statistics (Appendix D);
- Public Release and Data Usage Terms (Appendix E);
- Additional Results (Appendix F);
- Additional Analysis (Appendix G);
- Use of Large Language Models (LLMs) (Appendix H).

## A    DETAILS OF TAXONOMY

The comprehensive presentation of the full taxonomy of HumanPCR is shown in Figure 11.

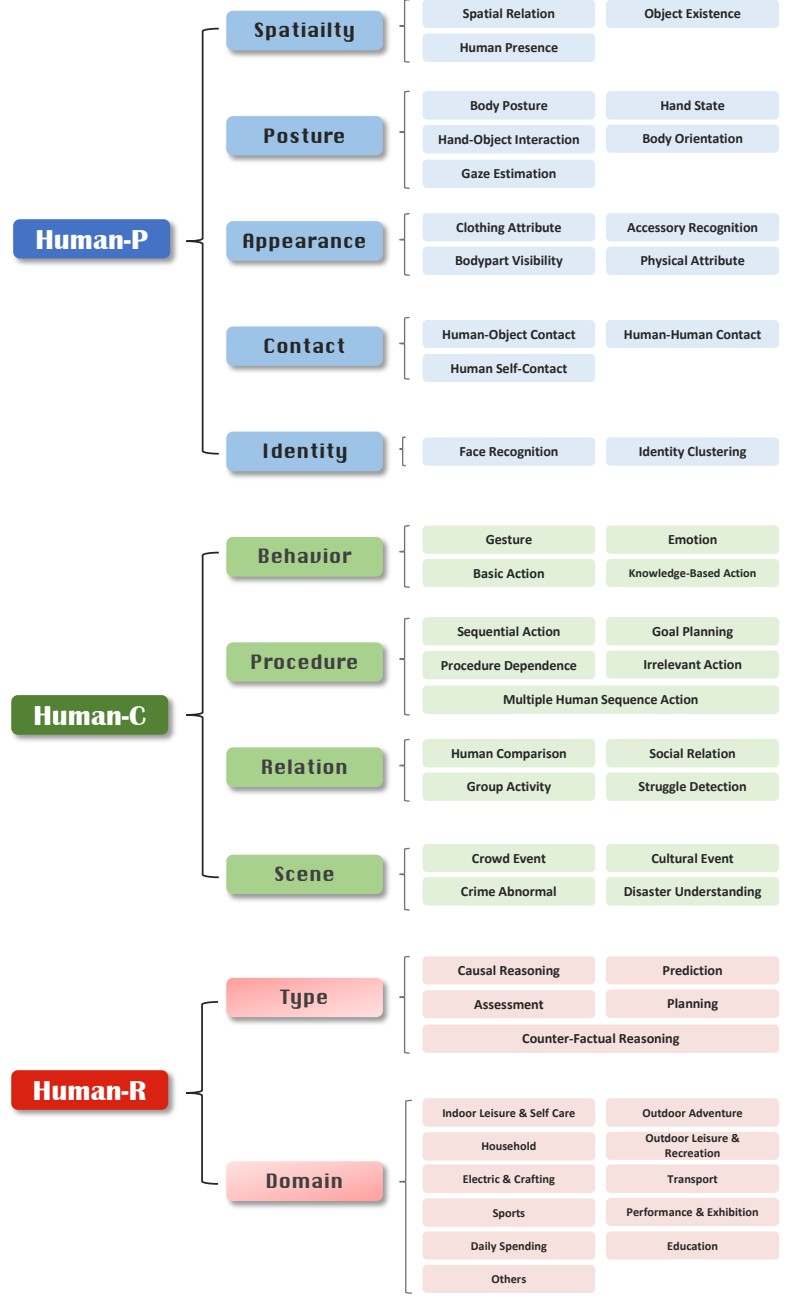

Figure 11: A comprehensive presentation of the full taxonomy.

## B    ANNOTATION SETUP

### B.1    ANNOTATION AND FILTERING PROMPTS

**Question Generation Prompts.**    To automatically generate high-quality questions, we employ two complementary strategies: LLM-based generation and template-based generation. In the LLM-based

pipeline, we feed GPT-4o with carefully crafted prompts augmented by comprehensive auxiliary information (see example prompts in Figure 12). In the template-based pipeline, we start from a set of seed templates and task the LLM with expanding them into a more diverse collection of question templates (examples appear in Figure 13).

```
### I will provide a caption about the spatial relationship of two
    object, the label of the spatial relationship. For 3 inputs, you
    will generate one diverse question for each input that can
    examine the following aspects:
Correctly answer the multiple-choice exam about recognizing the
    spatial relationsip of two objects in the image, which is
    described by the caption.

### As for your questions, you should ensure:
1. Output only question text.
2. Anyone can not obtain the answer just by reading the question
    text. Avoid implying answers in the questions.
3. Do not include specific visual details of the scene in the
    question.
4. Use plain text. Do not use Markdown format.
5. Make the questions concise, simple and straightforward. Do NOT
    add imaginary details to the questions.
6. Make the answer to the question point towards the spatial
    relation label of the two objects.
7. Do NOT disclose the spatial relation label in the questions. Do
    not say "caption" in questions.

### When announcing the question please label each question as '
    Question 1,2,3: [full question]'. Don't repeat the input.

### The input form for an image will be:
1.
Spatial Relation Caption: {caption}
Spatial Relation Label: {label}

2.
Spatial Relation Caption: {caption}
Spatial Relation Label: {label}

3.
Spatial Relation Caption: {caption}
Spatial Relation Label: {label}
```

Figure 12: Example prompt of LLM-based question generation used in the task `Spatial Relation`
.

**Option Generation Prompts.** To generate high-quality options, we decouple option generation from question generation. During option generation, we provide the LLM (GPT-4o) with the question, its correct answer, and supplementary context. The model is then instructed to produce options that are all pertinent to the question, with incorrect options crafted to be plausible yet wrong, thereby increasing the overall difficulty. An example prompt employed for option generation is illustrated in Figure 14.

**Blind Filtering Prompt.** After acquiring the automatically generated annotations, we eliminate any question that can be answered correctly without visual input. Specifically, we prompt GPT-4o to provide its optimal answer to each question in the absence of the corresponding image or video. Any question that is answered correctly across multiple runs—with the answer choices randomly

```
"What is the position of the person's hand in the picture?"
"How is the person's hand positioned in the image?"
"What is the state of the person's hand in the photo?"
"How is the hand of the person in the picture?"
"What condition is the person's hand in within the image?"
"In the picture, what is the status of the person's hand?"
"How does the person's hand appear in the photograph?"
"What is the arrangement of the person's hand in the image?"
"What is the specific condition of the person's hand in the picture?"
"How exactly is the person's hand positioned in the image?"
"What precise state is the person's hand in within the photo?"
"Can you describe the exact state of the person's hand in the
    picture?"
"What is the detailed posture of the person's hand in the image?"
"How is the person's hand specifically arranged in the photograph?"
"What particular position is the person's hand in within the image?"
"How would you describe the exact condition of the person's hand in
    the picture?"
"What is the exact gesture of the person's hand in the picture?"
"How is the person's hand specifically posed in the image?"
"What is the detailed form of the person's hand in the photo?"
"Can you describe the precise hand gesture of the person in the
    picture?"
"What is the specific shape of the person's hand in the image?"
"How exactly is the person's hand configured in the photograph?"
"What distinct position is the person's hand assuming in the image?"
"How would you characterize the specific hand posture of the person
    in the picture?"
```

Figure 13: Example templates of template-based question generation used in the task `Hand State`. The ambiguous references in questions were subsequently identified and corrected by annotators.

permuted each time—is subsequently discarded. The prompt employed for blind filtering is illustrated in Figure 15.

## B.2 ANNOTATION AND REVIEW INTERFACE

We designed personalized annotation and review interfaces to maximize annotation quality and streamline the entire workflow. The annotation interface guides each annotators to record the question, detailed reasoning steps, final answer, question category, and the relevant time interval. The review interface then requires reviewers to verify every annotated field and to assign both objective evaluations and subjective scores according to a comprehensive checklist. Finally, both interfaces include integrated messaging channels, enabling iterative feedback loops between annotators and reviewers and thus ensuring consistently high-quality annotations. A few snapshots of annotation and review interfaces is shown in Figure 16.

```
### I will provide a caption about the spatial relationship of two
    object, the label of the spatial relationship, and the
    questions, and 3 diverse questions that can examine the
    following aspects:
Correctly answer the multiple-choice exam about recognizing the
    spatial relationsip of two objects in the image, which is
    described by the caption.

### Based on the provided questions, I want you to create a
    difficult and diverse multiple-choice exam that tests the above
    aspects.

Each question should have 5 short answers, including 1 correnct
    answer and 4 wrong answers. Each answer option should reflect a
    reasonable understanding of a broadly similar but
    detail-different image.

The wrong answers should diverge from the correct ones only by
    fine-grained and subtle details that are easily mistaken.

### As for your answers, you should ensure:
1. Only one answer will be correct.
2. Answers are short and concise. Answers should not include
    irrelevant details that weren't queried.
3. Use plain text. Do not use Markdown format.
4. Make the answers concise, simple and straightforward. Do NOT add
    imaginary details to the answers.
5. Do NOT change the given questions.

### Print a question and then print each correct answer on a new
    line exactly as "Correct answer: [full answer]" Please print
    each wrong answer on a new line and print each wrong answer as
    "Wrong answer 1,2,3,4: [full answer]. Parse question-answer
    pairs by '\n\n'.

The input form will be:
1.
Spatial Relation Caption: CAPTION
Spatial Relation Label: LABEL
Questions: QUESTIONS

2.
Spatial Relation Caption: CAPTION
Spatial Relation Label: LABEL
Questions: QUESTIONS

3.
Spatial Relation Caption: CAPTION
Spatial Relation Label: LABEL
Questions: QUESTIONS
```

Figure 14: Example prompt of option generation used in the task Spatial Relation

```
You will be presented with a multiple-choice question, each with a
    fixed set of labeled options (e.g. A, B, C, D).
Your task:

1. You **must** choose exactly one of the provided --optionsA, B, C,
    etc.
2. Do **not** say "'theres not enough information," "no answer", or
    refuse.
3. If you genuinely cannot determine a "correct" answer, pick the
    option that seems **most plausible**.
4. Only output a option label, e.g.: "A".
```

Figure 15: The prompt employed for blind filtering.

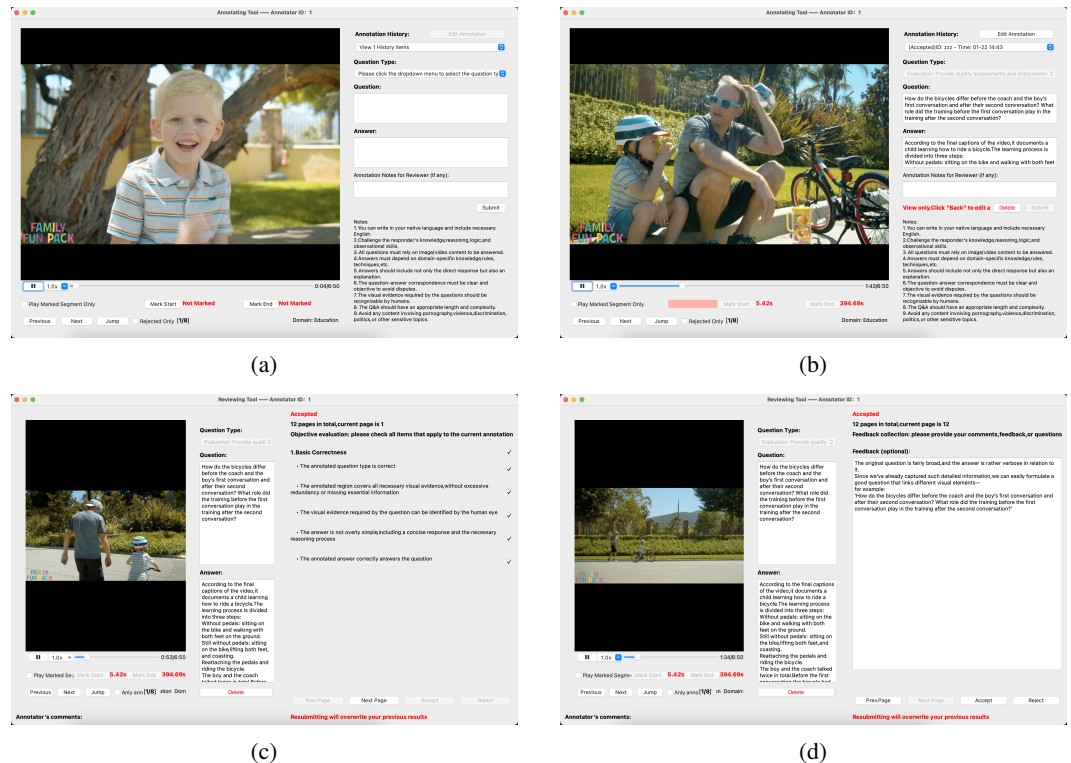

(a)

(b)

(c)

(d)

Figure 16: Snapshots of annotation and review interfaces.

## B.3 Detailed Annotation and Review Protocol for Human-R.

This subsection extensively describes the annotation and review processes employed in constructing Human-R, elaborating details beyond the main text.

Table 7: Example video tags of 11 human-related domains for Human-R.

| Domain | Tags |
|---|---|
| **Electric & Crafting** | Smart home devices, Kitchen appliances, Mobile device setup, Audio-visual equipment, Power tools usage, 3D printers, Smart thermostats, Electric scooters, Solar panel installation, Portable power stations |
| **Outdoor Leisure & Recreation** | Trail running, Paddleboarding fitness, Aqua aerobics, Slacklining, Outdoor yoga, Hot yoga, Balance and stability training, Kickboxing sessions, Animal flow workouts, Recreational cycling |
| **Outdoor Adventure** | Obstacle course training, Mountain biking, Marathon training plans, Ultra-trail running, Kayaking expeditions, Rock-climbing practice, Snowboarding sessions, Zip-line experiences, Caving basics, White-water rafting |
| **Household** | Wound cleaning and bandaging, Correct thermometer use, Home blood-pressure checks, Blood-glucose meter steps, Oral hygiene care, PPE donning methods, Nail disinfection, Skin antisepsis steps, Baby-bathing technique, Umbilical-cord care |
| **Sports** | HIIT workouts, Powerlifting programs, CrossFit sessions, Spinning classes, Strength-training routines, Cardio circuits, Kettlebell workouts, Rowing-machine drills, Plyometric exercises, Speed-and-agility drills |
| **Transport** | People in the subway, Passenger behavior on buses, Rush-hour metro crowds, Live bus footage, Commuters on trains, Night-bus activities, Airport-terminal scenes, Cycling commuters, Ferry passenger journeys, Public-transport safety tips |
| **Indoor Leisure & Self Care** | Yoga sessions, Pilates classes, Stretching routines, Home blood-pressure check, Meditation practice, Resistance-band workouts, Prenatal yoga, Postnatal fitness, Power yoga, Handstand training |
| **Daily Spending** | Supermarket shopping vlog, Online-shopping process, Restaurant dining vlog, Ordering food delivery, Thrift-store shopping, Paying bills in person, Buying electronics in store, Unboxing subscription boxes, Booking movie tickets online, Coffee-shop visits |
| **Performance & Exhibition** | Attending a concert vlog, Street artists on trains, Metro musical performances, Flash mobs on buses, Visiting a museum, Public art in stations, Art-workshop vlog, Open-mic night, Theatre-performance vlog, Dance-recital backstage |
| **Education** | Health-education workshop, Diabetes-education session, First-aid training demo, Speech-therapy practice, Clinical-trial walkthrough, Nutrition-counseling session, Community-health education, Occupational-health lesson, Travel-health briefing, CPR operation video |
| **Others** | Community health-center routine, Cancer-screening demo, Sleep-study procedure, Travel-health consultation, Language-therapy session, Geriatric patient care, Dietitian counseling, Pain-management therapy, Vaccination outreach, Sports-injury treatment |

### B.3.1 Annotation Process

**Step 1: Video Collection.** To ensure domain coverage and contextual diversity, videos were sourced from 11 human-related domains (as illustrated in Figure 7) by two main channels. (a) We leveraged curated egocentric and third-person activity datasets—Ego4D (Grauman et al., 2022),

EgoExo4D (Grauman et al., 2024), ActivityNet (Caba Heilbron et al., 2015), and MOMA (Luo et al., 2021)—and relied exclusively on their test or validation splits to avoid data leakage. (b) We collected internet videos from YouTube using domain-specific auto-generated tags. Clips under five minutes or containing low-quality or off-topic content were excluded. Only videos published within the past two years were considered.

**Step 2: Annotation Protocol.**   Visual reasoning involves extracting visual evidence across granularities and integrating them with world knowledge to perform inference. To ensure this, each question must involve at least two distinct visual evidences—defined as either specific attributes (e.g., orientation, action) or relationships (e.g., events, interactions)—and must invoke knowledge triggered by visual content. Trivial tasks like unconditioned counting or basic captioning are excluded.

To ensure diversity and evaluability, questions are created within one of five designated reasoning types. *Causal Reasoning* involves identifying cause-effect relations in the video, such as "Why did the person fall after stepping on the mat?"; *Prediction* involves anticipating what might happen next, e.g., "What will the person likely do after picking up the toolbox?"; *Counter-Factual Reasoning* asks about hypothetical alternatives, such as "What would have happened if the person had not pulled the lever?"; *Assessment* asks for judgment based on visual evidence and criteria, such as "Which athlete demonstrated better form during the lift?"; and *Planning* requires proposing a viable plan grounded in context, such as "What should the worker do next to safely continue the repair?"

**Step 3: Annotator Qualification and Tasks.**   Annotators were selected based on domain expertise. For specialized domains such as Education, Sports, Electric & Crafting, and Transport, annotators held relevant undergraduate degrees or had at least three years of experience. For general topics, annotators were computer science graduate students unaffiliated with the author team. All annotators underwent standardized training and trial tasks.

Each annotator was responsible for formulating open-ended questions that met the complexity criteria and fell under one of the designated reasoning types. They provided concise, direct answers and explicitly detailed the reasoning process, identifying essential visual evidence and associated knowledge. Annotators were allowed to flexibly choose relevant video segments to preserve contextual richness while avoiding oversimplification. To maintain fidelity in meaning, all annotations were first created in the annotators' native language and later translated professionally. Based on reviewer feedback, annotators were allowed to revise their submissions once.

### B.3.2   REVIEW AND QUALITY CONTROL PROCESS

**Step 4: Initial Review.**   Trained reviewers manually evaluated each submission for objectivity, factual correctness, conciseness, and reasoning complexity. To encourage proactively seeking the visual reasoning evidence, a focal point in this process was eliminating reference redundancy, where questions explicitly mention or imply the essential visual evidence, thus undermining the need for true reasoning. Instead, reviewers encouraged general references and annotated potential revision points with detailed feedback. Annotations that passed this stage without major issues advanced to the next round, while others were returned once for revision or rejected outright if irreparable.

**Step 5: Meta-Review.**   Meta-reviewers, typically senior researchers, conducted a second round of checks with a higher threshold for complexity and reasoning depth. They ensured each annotation incorporated at least two distinct pieces of visual evidence and required the integration of external knowledge triggered by visual content. Annotations relying solely on dominant cues or surface-level understanding were filtered out. The number of visual and proactive evidence per question was also counted and finalized after cross-validation.

**Outcome.**   This layered quality control pipeline, with stringent emphasis on complexity and reasoning integrity, resulted in a final acceptance rate below 20%. Due to the meticulous verification process, the average cost per question, including annotation and review, reached approximately $12. The outcome was a set of 442 high-quality reasoning questions, each crafted to challenge and benchmark advanced visual understanding capabilities.

### B.4 Annotator Recruitment in Human-R Construction

As detailed in Table 7, Human-R covers 11 diverse domains. In constructing this dataset, we prioritized *domain relevance and expertise* in annotator recruitment over a rigid "one-size-fits-all" standard, so that the questions and reasoning chains reflect realistic domain-specific characteristics rather than generic templates.

For **specialized domains** (e.g., *Sports, Electric & Crafting, Transport*), we recruited annotators with relevant practitioner experience or educational backgrounds. For instance, sports-related questions were assigned to fitness coaches or student athletes, while Electric & Crafting items were handled by annotators with engineering-related training or degrees. This ensured that both the question design and the associated CoT-style reasoning remained faithful to how domain experts would actually reason in these settings.

For **general domains** (e.g., *Household, Daily Spending*), we instead engaged high-quality annotators with strong performance in our qualification trials (i.e., high pass rates), without requiring formal professional credentials.

Overall, the dataset was constructed by a dedicated team of annotators recruited specifically for this project. We ensured that each domain had at least two annotators. Each annotator was assigned to domains aligned with their background and was required to pass qualification trials based on our annotation guidelines before contributing to the final dataset. This process helped ensure that domain expertise and annotation quality were systematically controlled across all 11 domains.

## C Evaluation Setup

### C.1 Configuration of Evaluated Models

Table 8 detail the configuration of each evaluated models on HumanPCR. Across all experiments, we use the default settings from the official implementation of each model to process vision input while the temperature is set to 1.0 and the maximum output length to 1,024 tokens, except for proprietary reasoning models (e.g., o3 and Gemini-2.5-pro-preview), for which the maximum output length is extended to 4,096 tokens to accommodate their extended CoT reasoning mechanisms. Additionally, for the Human-R evaluation, we tested all proprietary models and open-source models larger than 8B with frame numbers of 32, 64, and 128, and selected the configuration that yielded the best performance. All experiments are conducted using LMMs-Eval (Zhang et al., 2024a) on 8 Nvidia A100 GPUs.

For experiments of context extraction and RL-based thinking methods, we implemented their results by their official repository. Table 9 shows their configuration.

### C.2 Evaluation Prompt

We present the prompts in the evaluation in the following figures, for answering multiple-choice (Direct Answer in Figure 17 and CoT in Figure 18) and open-ended questions (Direct Answer in Figure 19 and CoT in Figure 20), respectively. The prompt for accuracy evaluation is presented in Figure 21.

### C.3 Reliability of LLM-based Judges

To ensure the robustness of our evaluation methodology, we tested multiple LLM judges and benchmarked their performance against human evaluation. For the human judgement, four annotators scored 4,000 model responses for per-instance accuracy and pairwise win-loss preference. The win-loss preference for LLM judges was determined by comparing their assigned scores across instances. Figure 10 in the main paper illustrates the evaluation results from four distinct judge models: o3-mini, gemini-2.5-flash, gpt-4o, and o3 and our findings are listed as below:

**High consistency in relative model rankings across all LLM judges** Performance orders of the evaluated models remain stable regardless of the judge employed, indicating that our evaluation outcomes are robust and not dependent on any single judge.

Table 8: Configurations of evaluated MLLMs on HumanPCR.

| Organization | Model | Release | Version | Input Frames (P&C/R) |
|---|---|---|---|---|
| *Open-source Models* | | | | |
| Rhymes | Aria | 2024-11 | `Aria-Chat` | 32/64 |
| Shanghai AI Lab | InternVL2-8b | 2024-6 | `InternVL2-8b` | 32/32 |
| | InternVL2.5-1B | 2024-11 | `InternVL2.5-1B` | 32 |
| | InternVL2.5-2B | 2024-11 | `InternVL2.5-2B` | 32 |
| | InternVL2.5-4B | 2024-11 | `InternVL2.5-4B` | 32 |
| | InternVL2.5-8B | 2024-11 | `InternVL2.5-8B` | 32 |
| | InternVL2.5-38B | 2024-11 | `InternVL2.5-38B` | 32/64 |
| | InternVL2.5-78B | 2024-11 | `InternVL2.5-78B` | 32/64 |
| | InternVL3-1B | 2025-4 | `InternVL3-1B` | 32 |
| | InternVL3-2B | 2025-4 | `InternVL3-2B` | 32 |
| | InternVL3-8B | 2025-4 | `InternVL3-8B` | 32/64 |
| | InternVL3-14B | 2025-4 | `InternVL3-14B` | 32/64 |
| | InternVL3-38B | 2025-4 | `InternVL3-38B` | 32/64 |
| | InternVL3-78B | 2025-4 | `InternVL3-78B` | 32 |
| Alibaba | Qwen2-VL-7B | 2024-8 | `Qwen2-VL-7B-Instruct` | 32 |
| | Qwen2.5-VL-7B | 2025-2 | `Qwen2.5-VL-7B-Instruct` | 32/64 |
| | Qwen2.5-VL-72B | 2025-2 | `Qwen2.5-VL-72B-Instruct` | 32 |
| lmms-lab | LLaVA-NeXT-Video-34B | 2024-5 | `LLaVA-NeXT-Video-34B` | 32 |
| | LLaVA-OneVision-7B | 2024-9 | `llava-onevision-qwen2-7b-ov-hf` | 32/128 |
| | LLaVA-OneVision-72B | 2024-9 | `llava-onevision-qwen2-72b-ov-hf` | 32 |
| | LLaVA-Video-7B | 2024-10 | `LLaVA-Video-7B-Qwen2` | 32/128 |
| | LLaVA-Video-72B | 2024-10 | `LLaVA-Video-72B-Qwen2` | 32 |
| OpenBMB | MiniCPM-V-2.6 | 2024-8 | `MiniCPM-V-2_6` | 32 |
| | MiniCPM-o-2.6 | 2025-1 | `MiniCPM-o-2_6` | 32 |
| Thu | Oryx-1.5-7B | 2024-10 | `Oryx-1.5-7B` | 32/64 |
| | Oryx-1.5-32B | 2024-10 | `Oryx-1.5-32B` | 32 |
| Nvidia | LongVILA-256 | 2024-12 | `qwen2-7b-longvila-256f` | 128 |
| | NVILA-8B | 2024-12 | `NVILA-8B` | 64 |
| *Proprietary Models* | | | | |
| OpenAI | GPT-4o | 2024-8 | `gpt-4o-2024-08-06` | 32/64 |
| | o4-mini | 2025-4 | `o4-mini-2025-04-16` | 32/64 |
| | o3 | 2025-4 | `o3-2025-04-16` | 64 |
| Google | Gemini-1.5-Flash | 2024-9 | `gemini-1.5-flash` | 32/64 |
| | Gemini-1.5-Pro | 2024-9 | `gemini-1.5-pro` | 32 |
| | Gemini-2.0-Flash | 2024-12 | `gemini-2.0-flash` | 32/64 |
| | Gemini-2.5-Flash | 2025-4 | `gemini-2.5-flash-preview-04-17` | 32/64 |
| | Gemini-2.5-Pro | 2025-3 | `gemini-2.5-pro-preview-03-25` | 64 |
| Anthropic | Claude-3.5-Sonnet-v2 | 2024-10 | `claude-3-5-sonnet-20241022` | 32/64 |
| | Claude-3.7-Sonnet | 2025-2 | `claude-3-7-sonnet-20250219` | 64 |
| ByteDance | Doubao-1.5-vision-pro | 2025-1 | `doubao-1.5-vision-pro-32k-250115` | 32/80 |
| xAI | Grok-2-Vision | 2024-12 | `grok-2-vision-1212` | 32/80 |

Question: {question}
A: {option_a}
B: {option_b}
C: {option_c}
D: {option_d}
E: {option_e}

Answer with the option's letter from the given choices directly.

Figure 17: Direct Answer prompt for multiple-choice questions.

**Strong correlation between LLM-based evaluations and human judgment** The Pearson correlation coefficients for accuracy are consistently high (0.640 to 0.689), and the alignment for pairwise win-loss preference is even stronger (0.832 to 0.872). We also found that providing judges with our

Table 9: Overview of evaluated context-extraction strategies and RL-based thinking models. The extraction granularity of the context of these strategies and all official code resources is also presented.

| Models/Methods | Baseline MLLM | Strategy | Granularity | Frame Cnt. | Repository URL |
|---|---|---|---|---|---|
| Video-RAG(Luo et al., 2024) | LLaVA–Video–7B | Memory&Retrieve | Frame | 64 | https://github.com/Leon1207/Video-RAG-master |
| AKS(Tang et al., 2025) | LLaVA–Video–7B | Frame Selection | Frame | 128 | https://github.com/ncTimTang/AKS |
| FRAG(Huang et al., 2025) | Qwen2.5-VL-7B | Frame Selection | Frame | 32 | https://github.com/NVlabs/FRAG |
| $T^*$(Ye et al., 2025) | Qwen2.5-VL-7B | Frame Selection | Frame | 32 | https://github.com/mll-lab-nu/TStar |
| PVC (Yang et al., 2024a) | InternVL2–8B | Down Pooling | Token | 512 | https://github.com/OpenGVLab/PVC |
| IXC2.5–OmniLive(Zhang et al., 2024b) | InternVL2–8B | Memory&Retrieve | Frame | 64 | https://github.com/InternLM/InternLM-XComposer |
| LongVU(Shen et al., 2024) | LLaVA–Video–7B | Token Selection | Frame & Token | 1 fps | https://github.com/Vision-CAIR/LongVU |
| Ola(Liu et al., 2025) | Qwen2.5-VL-7B | Down Pooling | Token | 64 | https://github.com/Ola-Omni/Ola |
| Dispider(Qian et al., 2025) | LLaVA–Video–7B | Memory&Retrieve | Frame | 512 | https://github.com/Mark12Ding/Dispider |
| AdaReTaKe(Wang et al., 2025b) | Qwen2.5-VL-7B | Token Selection | Token | 4 fps | https://github.com/SCZwangxiao/video-FlexReduc |
| FlexSelect(Zhang et al., 2025) | Qwen2.5-VL-7B | Token Selection | Token | 2 fps | https://github.com/yunzhuzhang0918/flexselect |
| Video–R1(Feng et al., 2025) | Qwen2.5-VL-7B | Thinking | - | 32 | https://github.com/tulerfeng/Video-R1 |
| Video–RFT(Feng et al., 2025) | Qwen2.5-VL-7B | Thinking | - | 32 | https://github.com/Liuziyu77/Visual-RFT |
| VideoChat–R1(Li et al., 2025a) | Qwen2.5-VL-7B | Thinking | - | 32 | https://github.com/OpenGVLab/VideoChat-R1 |

Question: {question}
A: {option_a}
B: {option_b}
C: {option_c}
D: {option_d}
E: {option_e}

Based on the given {modality}, select the best answer to the multiple-choice question by thinking step by step. Begin by explaining your reasoning process clearly. Conclude by stating the final answer using the following format: "FINAL ANSWER: \$LETTER" (without quotes), where \$LETTER is one of the option's letter from the given choices. Think step by step before answering.

Figure 18: CoT prompt for multiple-choice questions, partially adopted from previous works (Zhao et al., 2025; Yue et al., 2024). "modality" ranges in "image", "multiple images", and "video".

> Question: {question}
>
> Answer the question based on the given video. Do not generate any intermediate reasoning process. Directly output the final answer.

Figure 19: Direct Answer prompt for open-ended questions.

> Question: {question}
>
> Answer the question based on the given video step by step. Begin by explaining your reasoning process clearly. Conclude by stating the final answer using the following format: 'Therefore, the final answer is: "Answer: $ANSWER" (without quotes), where $ANSWER is the final answer of the question. Think step by step before answering. Below are frames uniformly sampled from the video."

Figure 20: CoT prompt for open-ended questions, partially adopted from previous works (Zhao et al., 2025; Yue et al., 2024).

annotated CoT as a reference significantly improved their alignment with human annotators (e.g., 0.642 vs. 0.689 for o3-mini W or W/o CoT on accuracy).

**Self-Bias Test.** We further conducted a Welch's t-test on whether a judge might favor models from its own family. The null hypothesis assumes that the score deviation over averages of other models of a judge for models from its own family is not significantly different from that for non-family models. With $\alpha = 0.05$, the test yields high p-values (e.g., p=0.864 for o3-mini judging o3; p=0.774 for gemini-2.5-flash on Gemini-family models), leading us to fail to reject the null hypothesis. This indicates very little statistical evidence of self-bias in our evaluation setting.

Collectively, these findings validate that in our setting, LLM judges serve as a reliable and robust alternative to human evaluation. We attribute this to the use of precise reference answers and detailed reasoning traces, combined with modern LLMs' strong instruction-following ability, which helps eliminate subjective noise and ambiguity in evaluation.

## C.4 EVALUATION SETTING OF HUMAN PERFORMANCE

We have benchmarked human performance using a subset of HumanPCR. For Human-P and Human-C, over 30 questions are sampled for each task while the evaluations were conducted by annotators who were not involved in the construction of the benchmark. For Human-R, accuracy was computed based on approximately 10% of Human-R questions completed by two graduate students in computer science who were not involved in the construction of the benchmark; during the annotation process, the annotators were allowed to consult a search engine for factual lookup or translation, but was instructed to refrain from locating the original source video or any accompanying ancillary materials by any means. The annotators received performance-contingent incentives: a correct answer paid twice the amount awarded for an incorrect one.

Evaluate **the accuracy score** of the model's answer and **whether its final answer is correct** by comparing it to the ground-truth answer provided for the given question.

You should first extract the final answer from the model's response, and then compare the extracted answer with the ground-truth answer to determine its accuracy. The final answer of the model does not need to match the ground-truth answer word-for-word. It should only be considered correct

1. if the final answer of the model demonstrates the consistent meaning or concept equivalent to the ground-truth answer.

2. if the final answer of the model meaningfully includes the ground-truth answer and does not introduce fundamentally different or unrelated meanings.

3. if the final answer of the model is meaningfully included within the ground-truth answer and only differs by omitting explanatory details instead of lacking key concepts.

Then, you should provide the accuracy score of the answer, which ranges from 0 to 4, based on both the correctness of the final answer and the accuracy of the reasoning process.

- If the final answer is incorrect, the score must be 0, 1, or 2:

    **0**: The final answer is incorrect, and the provided reasoning/evidence is either missing or entirely dissimilar to the groundtruth.

    **1**: The final answer is incorrect, but some visual details, reasoning steps, or evidence partially overlap with the groundtruth; however, most of the reasoning is incorrect.

    **2**: The final answer is incorrect, but the majority of the reasoning process, including key visual evidence and logical steps, aligns with the groundtruth, with only minor deviations causing an incorrect conclusion.

- If the final answer matches the groundtruth, the score must be 3 or 4:

    **3**: The final answer is correct, but the reasoning process or supporting evidence significantly differs from the groundtruth.

    **4**: The final answer is correct, and the reasoning process, including supporting evidence, closely aligns with the groundtruth without major inconsistencies.

Output your response in the following structured json object format:

```
{
  'extracted answer': // str value, the short final answer
      extracted from the 'models response, do not hallucinate one
      that is not present in the response,
  'correctness': // boolean value, True if the answer is correct,
      False otherwise,
  'score': // int value, overall assessment of accuracy of the
      model's answer
}
```

**Input Format**:
Question: {question}
Ground Truth Answer: {ground_truth}
Model Response to the Question: {model_response}

Figure 21: Evaluation prompt used for assessing the accuracy of open-ended questions.

# D    MORE DATA STATISTICS

Detailed video duration statistics across tasks of Human-P&C, domains and question types in Human-R are presented in Figure 22a, 22b, 22c, respectively. The average numbers of visual evidence across different domains and question types in Human-R are illustrated in Figure 23. The modality distribution of tasks is presented in Figure 24.

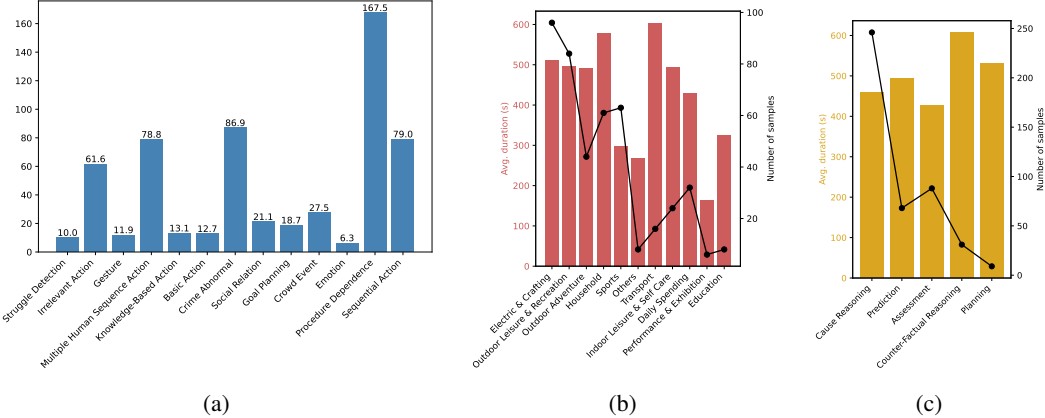

Figure 22: Statistics of video durations in HumanPCR. (a) The average durations for tasks in Human-P and Human-C. (b) The average durations(bar) and number of samples(line) for domains in Human-R. (c) The average durations(bar) and number of samples(line) for question types in Human-R.

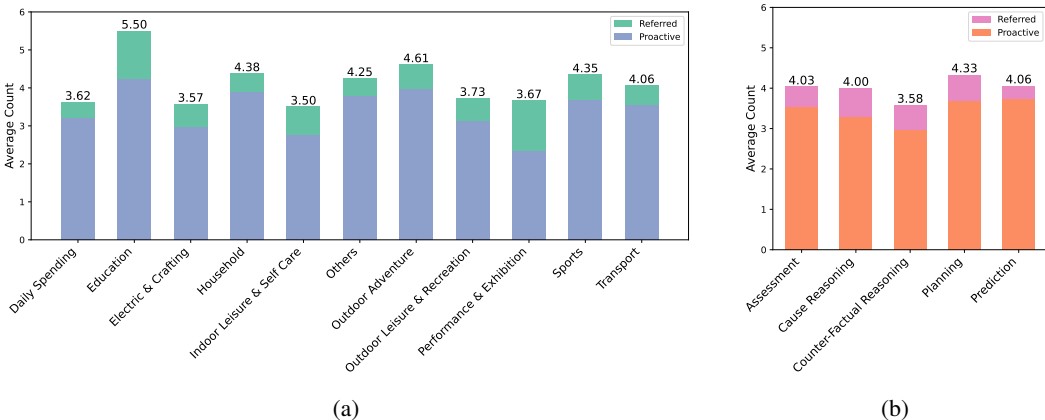

Figure 23: Average number of visual evidence in Human-R across (a) domains and (b) question types. The referred and proactive evidence are denoted by different colors.

# E    PUBLIC RELEASE AND DATA USAGE TERMS

To ensure minimal privacy and copyright issues while enabling open access for the research community, we have established the following Public Release and Data Usage Agreement (DUA). We publicly release all annotations created by us (and any data for which we are the rights holder) free of charge for academic research under our DUA, with the additional requirement that any use involving the original visual data must strictly comply with applicable privacy laws, institutional review policies, and obtain all necessary permissions. The DUA we formulated for the dataset open-source dataset is also presented in Figure 25.

**Scope and Hosting.** We do *not* redistribute raw third-party source media without explicit permission. For videos and images from public platforms (e.g., YouTube), we release only factual metadata (e.g.,

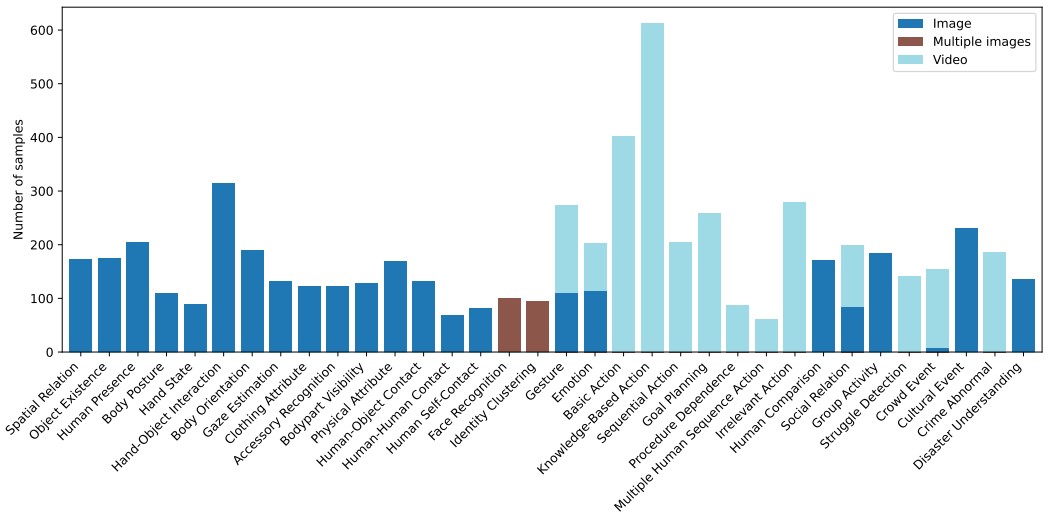

Figure 24: Distribution of samples of each modality across tasks.

video IDs, start/end timestamps for clips), and our task annotations. Source datasets that explicitly permit redistribution are hosted directly; those that prohibit it are provided via metadata, processing scripts, and links to original sources so users assemble content themselves.

**Access Control and Backup Channel.** A controlled (backup) access channel requires explicit agreement to (i) all upstream dataset licenses, (ii) a non-redistribution commitment, and (iii) our Data Use Agreement (DUA). Users must request access and may have it revoked upon breach.

**Data Use Agreement.** The DUA presented in Figure 25 forbids: re-identification, biometric template construction, surveillance or tracking applications, discriminatory or harmful usage, circumvention of platform Terms of Service, and repackaging or further redistribution of third-party content. Data are for evaluation/research; we do not train models on the benchmark annotations.

**Privacy Minimization.** Our annotation guidelines have excluded personal names, private locations, and sensitive attributes; quality control removed residual high-risk items. No biometric embeddings or identifiers are released. If a source video is deleted or privatized, its entry becomes unusable.

**Copyright Requests.** Rights holders may request modification or removal of hosted data at any time; validated requests are processed promptly and reflected in the next version.

**Disclaimer.** The benchmark is provided "as is" without warranty; users bear responsibility for legal and ethical compliance. Continued use constitutes acceptance of updated terms.

```
HumanPCR Data Use Agreement (DUA)

1. Content Hosting and Download
   The dataset only contains metadata (e.g., public video IDs and
       timestamps) for content sourced from public platforms.
   We do not host or distribute raw video files.
   Users are responsible for using the original videos themselves,
       following the Terms of Service of the source platforms (e.g.,
       YouTube).

2. License Compliance for Hosted Data
   For any datasets or files directly hosted by us, by accessing or
       using HumanPCR you automatically agree to comply with the
       original licenses and usage restrictions of those datasets.

3. Usage Restrictions

- HumanPCR is provided for **non-commercial, academic research
    purposes** only.
- You are **strictly prohibited** from using the data for biometric
    identification, tracking, or surveillance technologies.
- You must not attempt to re-identify, de-anonymize, or contact any
    individuals depicted in the images or videos.
- The data may **not** be used to discriminate against, harass, or
    negatively profile individuals or groups.
- Without prior written approval, you may not redistribute, publish,
    or disseminate the 'datasets metadata, including video and image
    data, in whole or in part; task annotations we created may be
    redistributed for non-commercial use with attribution, provided
    you link to this DUA and indicate any changes.
- Attribution requirement: When using HumanPCR metadata or our task
    annotations, you must give appropriate credit, provide a link to
    this DUA, and indicate if changes were made.
- No additional restrictions: You may not apply legal terms or
    technological measures that legally restrict others from doing
    anything the DUA permits.

4. Copyright and Removal Requests
   All copyrights remain with the original content owners.
   If you are a rights holder and believe there is an issue with any
       content referenced in HumanPCR, please contact us.
   We will promptly review and remove or update the relevant content
       as requested.

By accessing or using HumanPCR, you acknowledge and agree to comply
    with the above terms.
```

Figure 25: HumanPCR Data Use Agreement (DUA). This document ensures that the dataset is distributed responsibly for non-commercial academic research purposes only.

## F   MORE RESULTS

### F.1   FULL RESULTS ON HUMANPCR

The per-task accuracies of the evaluated model are presented in Table 10.

Table 10: Full results of evaluated models on HumanPRC for each task. The header of each dimension (e.g., Spa. for Spatiality) and each task (e.g., S.R. for Spatial Relation) are abbreviated for brevity.

```
                          Perception                                                              Comprehension            Open
Models         Spa.          Pos.          App.      Con.  Ide.  Beh.        Pro.                 Rel.            Sce.      Rea.
               S.R. OE. H.P. B.P. H.S. H.O.I. B.O. G.E. C.A. A.R.  B.V. H.P.A. H.O.C. H.H.C. H.S.C.  FR. I.C. G.U. E.R. B.A. K.B.A. S.A. G.P. P.D. MHSA.  I.A. H.C. S.R1. G.A. S.D.  C.E. CE1. C.AI. D.U.  Rea.

Open-source Models
Aria                  69.3685.6382.3546.7942.70 49.2126.3233.3972.1383.6133.59 55.95 29.41 28.0572.00 81.0540.1548.2863.34 43.6344.1222.3942.53 45.0052.5261.1851.7666.6720.71 36.3675.22 54.0555.56 28.96
LongVILA-256          67.8281.1478.9238.5325.84 46.0325.1323.6670.4990.9835.16 56.47 47.73 29.85 39.0258.00 44.3936.7618.1527.59 21.6755.4051.7656.2872.1324.29 33.7768.70 54.0556.30 21.49
NVILA-8B              61.4982.8684.8036.7025.84 48.2528.8037.4070.4994.2637.50 55.88 47.73 29.85 50.0058.00 37.8935.0447.2956.36 28.3347.4854.1253.7774.8620.00 32.4774.35 54.0551.85 22.17
MiniCPM-V-2.6         63.2280.5778.4345.8731.46 50.1627.2329.7758.2087.7032.81 52.35 40.91 32.84 32.9362.00 78.9541.2453.6954.61 26.6754.3258.2450.2570.4922.86 39.6170.00 53.5155.56 16.74
MiniCPM-o-2.6         74.1482.2983.8247.7132.58 54.2934.0350.3868.0390.1634.38 61.76 44.70 28.36 37.8072.00 70.5336.1353.2060.35 31.6742.8162.3556.7873.2223.57 39.6175.22 51.8954.81 21.04
LLaVA-Video-7B        64.1681.6178.4340.3731.46 48.8932.1132.8259.0288.5232.81 51.79 40.15 30.88 26.8367.00 60.0038.6947.7858.85 30.0051.0857.0657.7968.3115.00 36.3669.13 49.7351.85 27.60
LLaVA-Video-72B       67.2482.2980.3952.2937.08 53.0233.5138.9383.6190.1642.97 70.00 62.12 34.33 73.1762.00 65.2644.5351.2365.34 36.6750.7261.7661.3172.1326.43 46.1079.13 62.1656.30 28.28
LLaVA-OneVision-7B    64.1683.9185.7835.7831.46 52.7034.2127.4865.5788.5231.25 54.76 43.94 35.29 23.1764.00 45.2643.4344.3360.60 30.0050.3662.9453.7774.3218.57 31.8273.04 49.7349.63 22.85
LLaVA-OneVision-72B   75.7286.2185.7854.1337.08 54.9234.2143.5183.6191.8035.94 70.24 64.39 38.24 70.7366.00 77.8946.3550.2564.09 38.3349.2871.1860.3079.7821.43 46.1080.43 63.2456.30 27.60
LLaVA-NeXT-Video-34B  46.5569.1462.7540.3723.60 35.5630.3720.6157.3882.7928.91 48.82 40.15 26.87 42.6859.00 30.5332.8539.4147.63 31.7126.9614.2920.69 30.0052.8848.2458.2960.6622.86 32.4764.35 43.2243.70 15.16
Oryx-1.5-7B           61.2782.1880.3943.1234.83 48.8925.7929.7767.2190.1636.72 55.36 42.42 33.82 51.2251.00 39.3835.7742.8658.85 23.3352.5255.2954.7772.1325.71 34.4270.87 50.8140.74 22.17
Oryx-1.5-32B          76.3085.6384.3153.2129.21 54.6033.6833.5975.4192.6236.72 54.76 59.85 41.18 42.6850.00 45.1035.7819.3171.26 41.6755.0467.6560.3071.5818.57 46.7577.83 57.3048.15 28.51
Qwen2-VL-7B-Instruct  65.9078.1680.3933.9414.61 47.3030.5324.4365.5790.1628.12 55.95 47.73 27.94 31.7150.00 48.3738.7318.5332.18 25.0049.2854.7156.2868.8524.29 31.8279.57 54.0548.15 26.24
Qwen2.5-VL-7B         70.1184.0081.8648.6239.33 49.8433.5139.6965.5790.1635.94 57.65 46.21 23.88 21.9550.00 46.1840.2015.8329.89 26.6750.7259.4156.2868.3123.57 37.6678.26 54.0553.33 26.24
Qwen2.5-VL-72B        78.7482.2985.2961.4746.07 53.3338.7450.3878.6990.9844.53 60.59 62.12 44.78 24.3950.00 55.2854.4127.4159.77 38.3352.5265.2959.8072.1315.71 51.3081.74 65.4159.26 34.39
InternVL2-8B          67.0580.4678.4332.1128.09 44.7626.8435.8863.1185.2534.38 55.95 46.97 29.41 35.3770.00 58.9536.1341.3853.12 28.3346.4058.8255.2872.6821.43 37.6668.26 52.4351.85 17.65
InternVL2.5-1B        38.5173.1474.5136.7029.21 44.7625.6526.7258.2082.7933.59 41.18 50.00 26.87 34.1555.00 32.6327.3732.0250.62 38.5429.4116.60 9.20 23.3352.1648.2443.7261.2029.29 34.4249.13 42.1632.59 11.31
InternVL2.5-2B        51.1580.0078.4337.6142.70 47.9427.7529.0156.5686.8932.81 44.12 23.40 34.33 25.6162.00 36.8434.6740.8953.00 36.2640.9917.7613.79 18.3350.3645.8848.7429.9434.29 29.9460.87 43.2445.19 22.39
InternVL2.5-4B        63.2284.0079.4141.2837.08 50.7930.3736.6466.3992.6234.38 42.55 42.55 26.87 39.0265.00 66.3237.5952.2261.39 44.3940.0916.2234.48 28.3350.7261.1853.7736.9424.29 36.9466.09 51.8948.89 29.41
InternVL2.5-8B        74.1482.2981.8647.7131.46 54.2932.9841.9863.9386.8931.25 59.41 44.70 37.31 28.0573.00 77.8941.2446.8062.34 43.7440.6919.6955.17 33.3346.7665.8853.7774.3215.00 37.6671.74 59.4657.04 23.53
InternVL2.5-38B       78.7488.0086.2751.3839.33 55.5638.2241.2277.0592.6239.84 65.29 70.45 41.79 37.8077.00 91.5847.0845.8166.08 56.5959.8032.0570.11 53.3363.6772.9458.7971.0415.00 50.6580.00 60.5461.48 35.97
InternVL2.5-78B       82.0885.0687.2558.7237.08 56.5133.6837.4081.1590.1639.84 66.07 62.88 47.06 42.6873.00 89.4747.8153.2070.32 58.8258.8231.6678.16 41.6759.3574.7161.3176.5016.43 51.9580.87 66.4961.48 33.94

Proprietary Models
InternVL3-1B          42.2076.4472.0626.6132.58 44.7625.2626.7253.2886.8931.25 47.02 40.91 22.06 30.4957.00 31.5829.2036.9549.88 38.5429.4116.60 9.20 35.0047.8447.6549.2562.8428.5734.4249.13 38.3842.2211.99
InternVL3-2B          60.6980.4680.3934.8626.97 46.3528.4232.8250.8287.7032.03 41.67 23.53 19.5166.00 46.3235.4049.2656.36 40.6935.7825.4829.89 25.0046.0452.9450.7571.0432.8639.6167.39 48.6549.6320.36
InternVL3-8B          73.9985.6385.2939.9435.96 58.4136.8432.0676.2392.6239.84 60.12 53.79 27.94 32.9372.00 73.6843.0752.2264.84 45.4248.5325.1047.13 35.0048.5669.4156.2874.3220.00 40.9176.09 61.0858.52 31.45
InternVL3-14B         78.6183.3381.8642.2043.82 52.0641.5848.0972.9589.3442.19 67.26 62.12 39.71 62.2069.00 91.5844.8952.7166.33 51.1448.5331.6655.17 48.3351.4476.4761.8176.5015.71 53.9077.83 62.1662.22 31.00
InternVL3-38B         78.6187.3688.2452.2947.19 55.5641.5851.1582.7991.8033.59 68.45 67.42 47.06 59.7666.00 95.7952.1945.3273.82 60.1352.9442.0879.31 45.0057.5576.4765.3379.2315.00 59.0980.87 64.8659.26 35.75
InternVL3-78B         81.5087.9390.2054.1340.45 58.7334.7444.2786.8994.2642.19 70.24 65.91 43.90 76.00 96.8447.8150.7472.82 60.4654.9042.0873.56 41.6759.3573.5363.8280.3321.43 55.1982.61 77.8465.19 37.56
Doubao-1.5-vision-pro 64.1679.3174.0243.1222.47 49.2131.0538.9380.3390.9836.72 60.71 55.30 44.12 13.4185.00 71.5843.8035.4763.09 50.0061.5143.5356.2867.7616.43 35.7116.96 54.5947.41 32.81
Grok-2-Vision         41.0470.6961.7634.8625.84 37.7818.9537.4067.2179.5137.50 45.83 58.33 45.59 19.5172.00 28.4240.1540.8957.61 40.0058.2737.0652.2661.2018.57 40.2673.48 62.7049.63 36.20
Claude-3.5-Sonnet-v2  61.8572.9969.1242.2028.09 53.3330.5345.0462.3081.9731.25 61.90 64.39 29.41 40.2482.00 50.5346.7245.8163.59 40.0063.3148.8260.8068.8517.86 36.3683.48 64.3252.59 39.59
Gemini-1.5-Flash      39.6665.5259.8038.5338.20 38.1028.8048.0953.2873.7737.50 50.59 42.42 35.29 20.7371.00 37.8939.4245.3260.60 36.6762.2737.0653.2763.9324.29 31.1777.83 56.7641.48 35.97
Gemini-1.5-Pro        57.8072.9969.6142.2048.31 51.7533.1652.6758.2075.4132.81 57.74 57.58 39.71 20.7377.00 61.0546.3543.8466.33 30.0068.7140.0059.8069.4031.43 39.6180.00 71.3556.30 40.05
Gemini-2.0-Flash      72.9979.3176.9638.5351.69 56.8334.5555.7371.3192.6230.47 60.59 60.61 50.00 46.3483.00 63.1648.1844.3367.33 28.3375.1862.3560.8074.3220.00 40.2684.35 70.2748.15 38.01
Gemini-2.5-Flash      80.3583.3382.3543.1256.18 56.1938.4246.5682.7994.2637.50 63.10 66.67 48.53 32.9387.00 100.0051.4648.2864.84 30.0061.8772.3559.8079.7817.14 48.0583.48 73.5145.19 43.44
GPT-4o                60.6978.1671.5742.2039.33 55.5632.1133.5961.4886.0732.03 42.26 46.97 36.76 17.0746.00 24.2151.0938.9264.09 33.3354.3247.6555.7863.3928.57 35.0685.22 63.2457.04 41.40
o4-mini               77.4682.7681.8650.4651.69 57.7850.0055.7378.6994.2648.44 66.07 62.12 36.76 24.3976.00 95.7963.8753.6969.83 41.6775.9081.7660.3082.5122.14 54.5585.65 68.1155.56 53.39
```

Table 11: Uncertainty Reporting. The average accuracy, standard deviation, and 95% confidence intervals for Qwen2.5-VL-72B (Bai et al., 2025) and GPT-4o (Hurst et al., 2024).

| | Multi-Choice | | | | | | | | | | | | Open |
|---|---|---|---|---|---|---|---|---|---|---|---|---|---|
| | Human-P | | | | | | Human-C | | | | | | Human-R |
| | Spa. | Pos. | App. | Con. | Ide. | avg. | Beh. | Pro. | Rel. | Sce. | avg. | Acc. | Acc. |
| *Qwen2.5-VL-72B* | | | | | | | | | | | | | |
| Mean | 81.72 | 49.96 | 67.13 | 42.69 | 41.21 | 57.90 | 55.21 | 47.96 | 51.50 | 61.93 | 54.75 | 56.29 | 34.31 |
| Variance | 0.15 | 0.01 | 1.90 | 0.99 | 0.06 | 0.01 | 0.24 | 1.62 | 2.26 | 4.73 | 0.06 | 0.01 | 0.32 |
| 95% CI Lower | 80.75 | 49.79 | 63.71 | 40.22 | 40.58 | 57.63 | 54.00 | 44.80 | 47.76 | 56.52 | 54.14 | 56.09 | 32.90 |
| 95% CI Upper | 82.69 | 50.12 | 70.55 | 45.16 | 41.84 | 58.18 | 56.42 | 51.11 | 55.23 | 67.33 | 55.37 | 56.49 | 35.73 |
| *GPT-4o* | | | | | | | | | | | | | |
| Mean | 70.63 | 42.25 | 56.09 | 32.72 | 40.51 | 48.62 | 52.05 | 41.28 | 50.05 | 60.71 | 50.45 | 49.54 | 41.02 |
| Variance | 0.34 | 2.29 | 0.32 | 2.21 | 23.53 | 1.11 | 0.02 | 4.17 | 1.12 | 0.42 | 0.94 | 1.02 | 0.12 |
| 95% CI Lower | 69.18 | 38.50 | 54.68 | 29.02 | 28.46 | 46.01 | 51.71 | 36.21 | 47.42 | 59.10 | 48.04 | 47.03 | 40.16 |
| 95% CI Upper | 72.07 | 46.01 | 57.50 | 36.41 | 52.56 | 51.24 | 52.40 | 46.35 | 52.68 | 62.31 | 52.86 | 52.05 | 41.88 |

## F.2 RESULTS ON HUMAN-R ACROSS QUESTION TYPES AND DOMAINS

The results of Human-R broken down by question types and video source domains are presented in Figure 26.

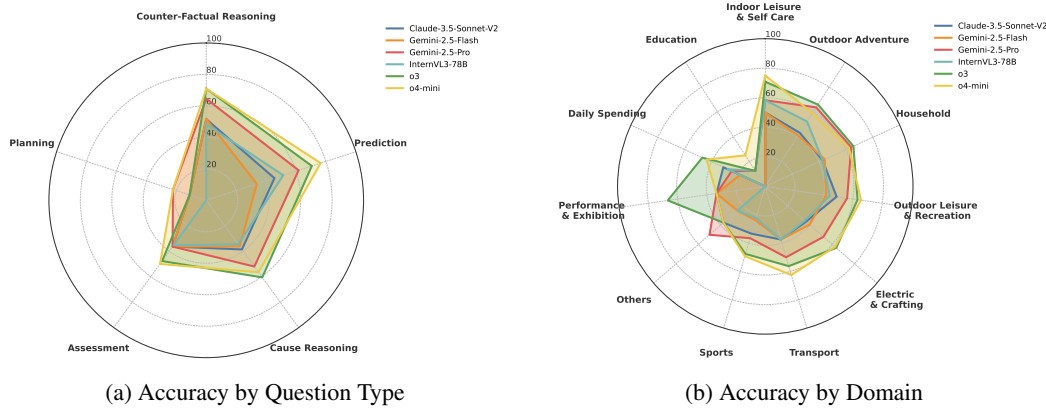

(a) Accuracy by Question Type       (b) Accuracy by Domain

Figure 26: Accuracy comparison on Human-R across (a) different question types and (b) video source domains. Most models struggle with Planning-type reasoning, while Education is the most challenging video domain.

## F.3 UNCERTAINTY REPORTING VIA REPEATED RUNS

For representative models, we run each experiment three times and report, for every level and dimension, the average accuracy, standard deviation, and 95% confidence intervals. This provides explicit uncertainty estimates to complement single-point metrics as shown in Tbale 11.

## F.4 COMPUTATIONAL RESOURCES AND RUNTIME DETAILS

To improve reproducibility and transparency, we report wall-clock runtimes and memory configurations across representative models and hardware setups. Our primary infrastructure consists of 8× NVIDIA A100 (80GB) GPUs. For models with around 7B paprameters (e.g., InternVL3-8B) under data parallelism on 8×A100, Human-P and Human-C require approximately 30 minutes in total, corresponding to 4 GPUh, while Human-R requires approximately 15 minutes, corresponding to 2 GPUh. For larger InternVL3 variants, we run the full evaluation suite with the following configurations and costs: InternVL3-14B on 1×A100 takes 6 hours; InternVL3-38B using model parallelism on 2×A100 takes approximately 13 hours; and InternVL3-78B on 3×A100 takes approximately 16 hours.

Table 12: Results of test-time scaling strategies on Human-R. Open-source and proprietary thinking models are both included. $M$ represents the magnitude of the strategies. Specifically, the BoN strategy uses $2^M$ candidates, while the Self-Refine strategy performs $M$ iterations.

| Model | Method | Reward Model | Direct | M=0 (CoT) | M=1 | M=2 | M=3 |
|---|---|---|---|---|---|---|---|
| **Test-Time Compute: Prompt Procedure** | | | | | | | |
| Gemini-2.0-Flash | BoN | Gemini-2.0-Flash (Self) | | | 36.88 | 38.01 | 38.69 |
| | BoN | Gemini-2.5-Flash | 32.13 | 36.43 | 37.55 | 37.10 | **41.86** |
| | BoN | o4-mini | | | 42.53 | 40.05 | 40.05 |
| | Self-Refine | - | | | 41.40 | 36.65 | 36.65 |
| GPT-4o | BoN | GPT-4o (Self) | | | 43.21 | 45.70 | 45.93 |
| | BoN | Gemini-2.5-Flash | 32.81 | 41.40 | 44.57 | 45.02 | **46.38** |
| | BoN | o4-mini | | | 44.11 | 46.38 | 45.02 |
| | Self-Refine | - | | | 39.59 | 39.82 | 40.95 |
| InternVL3-78B | BoN | InternVL3-78B (Self) | | | 32.35 | 35.06 | 34.16 |
| | BoN | Gemini-2.5-Flash | | | 36.65 | 40.95 | 38.91 |
| | BoN | o4-mini | **35.52** | 37.56 | 38.68 | 41.40 | **42.30** |
| | BoN | GPT-4o | | | 32.8 | 34.6 | 35.29 |
| | Self-Refine | - | | | 33.71 | 35.07 | 34.62 |
| **Proprietary Thinking Model** | | | | | | | |
| Gemini-2.5-Pro-preview(Comanici et al., 2025) | - | - | - | 51.13 | | - | |
| Claude-3.7-sonnet(Anthropic, 2025) | - | - | - | 40.50 | | - | |
| o3(OpenAI, 2025) | - | - | - | **59.28** | | - | |

# G ADDITIONAL ANALYSIS

## G.1 IMPACT OF MODEL SIZE.

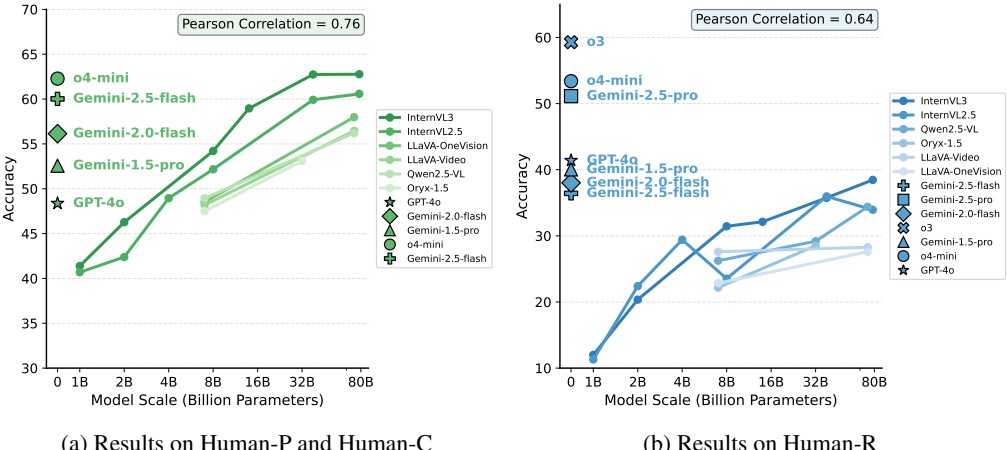

(a) Results on Human-P and Human-C          (b) Results on Human-R

Figure 27: The relationship between model size and macro-average accuracy.

In Table 3, scaling model size consistently improves accuracy across MLLMs series. As Figure 27 further reveals, performance of InternVL2.5 and InternVL3 suggests that complex reasoning tasks are more sensitive to model size scaling, which brings steady improvements, while performance on Human-P and Human-C plateaus beyond approximately 38B parameters. We hypothesize that complex tasks more effectively evaluate the fundamental capabilities of the models, such as world knowledge and logical inference.

## G.2 FULL RESULTS OF TEST-TIME SCALING STRATEGIES ON HUMAN-R.

Table 12 show the effect of test-time scaling strategies beyond vanilla CoT on Human-R. Best-of-N (BoN) delivers over 5% gains on all 3 models with incremental improvements from stronger reward models or more candidates. Self-Refine merely offers marginal gains and degrades with more iterations. Overall, test-time scaling strategies are generally effective.

## G.3 Reasoning Error Type Analysis

This section presents a structured analysis of error types based on a comprehensive categorization of model failures from top-performing models. The error type distribution is derived from the average error breakdown of the results from various settings, including o3, o3 (8f), GPT-4o, InternVL3-78B, Gemini-2.5-Pro-Preview, Gemini-2.0-Flash, and Gemini-2.0-Flash (1fps/1200f), computed over a representative set of 200 randomly sampled questions mentioned in the main text.

**Visual Element Extraction Error (37.42%).** This emerges as the most frequent failure type, highlighting fundamental difficulties in accurately identifying the necessary visual information from video contexts. This dominant category signifies that models often struggle to "see" the crucial components required for sound reasoning, either by failing to detect key elements or by mistakenly incorporating extraneous ones. This category is further broken down:

- **Missing Proactive Evidences (21.87% of total errors):** This is the largest sub-category, where models overlook essential visual cues not explicitly mentioned but to be extracted in the broader context. These include understanding the temporal order(e.g., misidentification of repeated events and incorrect verification of completed steps), concurrency (e.g., inferring a person's state and actions from others' reactions and changes in scenes), and general causes and consequences. Such errors reveal a fundamental weakness in holistic visual interpretation, indicating that models are not adept at spontaneously modeling context inner connections and structures.

- **Missing Referred Evidences (9.22% of total errors):** In these instances, models fail to identify visual elements that are explicitly mentioned or pointed to in the question. This challenge appears particularly in fine-grained actions (e.g., very brief actions or background events) and ambiguous references (e.g., multiple similar candidates to references with only one correct match), even when clear guidance is given. This highlights the need for MLLMs to expand long context and more efficient context encoding.

- **Involving Irrelevant Evidences (6.32% of total errors):** This subtype occurs when models incorporate visual details that are extraneous or unimportant to the query at hand. This inclusion of irrelevant information can derail the reasoning process by introducing noise, creating confusion, or leading to flawed inferences.

**Knowledge / Common-sense Error (36.63%).** This represents the second most significant hurdle, occurring when the model correctly perceives relevant visual elements but lacks the factual knowledge (e.g., confusing distance and time in speed calculation), domain understanding (e.g., unaware that a draw leads to a penalty shootout and thus unable to provide the correct score), or common sense (e.g., failing to explain that water cannot be poured because it has frozen due to low temperatures) needed to infer accurately and reach correct conclusions. This substantial percentage highlights that models' visual understanding is often weakly grounded in real-world elements or the logical principles required to derive broader implications or factual knowledge. This underscores the need for MLLMs to better adapt to expert human-domain data and uphold faithfulness during decoding.

**Perception Error (13.04%).** This occurs when the model explicitly misinterprets or incorrectly recognizes visual evidence that it has managed to extract, including misperception of visual attributes (e.g., incorrect classification of swimming styles), visual hallucinations, and confusion about attribute ownership(e.g., confusing subtitles from different video chapters). This type of error points to deficiencies in the model's core visual foundational capabilities.

**Refusal / Incomplete Answer (7.51%).** This captures instances where the model either explicitly refuses to answer or provides partial, uncertain responses. Beyond built-in safety policies, we observe that models also exhibit selective answering behavior—often refusing to respond due to lack of evidence, especially when few frames are provided. Another cause of partial answers is the model falling into repetition or prematurely ending its output during long reasoning processes.

**Question-understanding Error (5.4%).** captures failures caused by misinterpreting the query's intent or its constraints. For instance, models may provide a generic video description instead of answering the question, or ignore required answer or reference formats—e.g., failing to use ordinal

terms to identify actions, or giving only an answer when a rationale is explicitly requested. These errors highlight the model's fragility in instruction following and constrained decoding, indicating a need for tighter alignment between visual understanding, reasoning, and structured output formatting.

### G.4 QUALITATIVE EXAMPLES OF REASONING ERRORS

We investigate error cases of Human-R, illustrated in Figure 28, 29, 30, 31, including detailed reasoning process and error type analysis of each model.

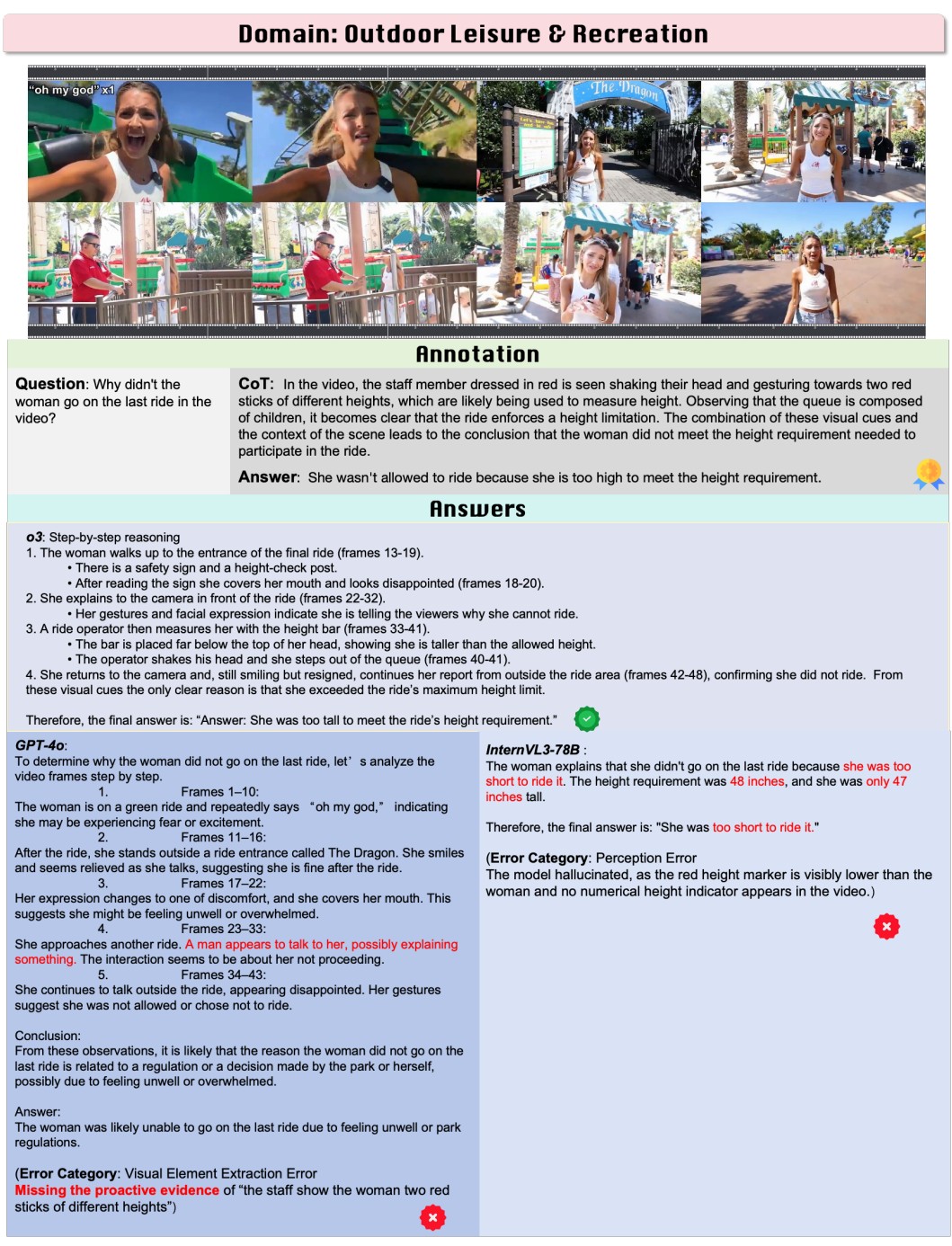

Figure 28: An annotation example of Human-R in the domain "Outdoor Leisure & Recreation" and its error analysis.

**Domain: Electric & Crafting**

**Annotation**

**Question**: How many of the tools introduced in the video are used for turning nuts?Refer to them in the order in which they appear in the video, such as "the first one.

**CoT**: The video clearly demonstrates multiple tools, each with a distinct function, and through visual observation and comparative analysis, it is evident that only two of these tools are designed for turning nuts—specifically, the second and the fourth tools shown. The sequence of tool usage further confirms this: the first is a DeWalt gyroscopic inline screwdriver, used for driving screws; the second is a Tempest oil-filter torque wrench, a ratcheting box wrench designed to fit and turn hex components like oil filters; the third is a pair of Snap-on cutting pliers, used for cutting safety wire; the fourth is a Knipex pliers-wrench, an adjustable smooth-jaw wrench made for gripping and turning hex nuts; and the fifth is a set of Vampliers, intended for gripping damaged screw heads rather than standard nut-turning.

**Answer**: Two, the second and the fourth tools are used for turning nuts.

**Answers**

**Gemini-2.5-Pro-Preview**:
**Step 1: Identify the tools introduced in the video.**
 Observe the frames to see which tools are presented individually by the speaker. The tools are laid out on the table in frames 1 and 2. The speaker then picks up and discusses several tools one by one.
 *  The first tool discussed is the yellow DeWalt power screwdriver (Frames 6-10).
 *  The second tool discussed is the socket wrench/ratchet, identified as a Tempest Oil Filter Torque Wrench (Frames 11-15).
 *  The third tool discussed is the small red-handled tool, which appears to be wire cutters or strippers (Frames 16-19).
 *  The fourth tool discussed is the Knipex Pliers Wrench (Frames 20-24).
 *  The fifth tool discussed is the pair of combination pliers (Frames 25-29).

*   **Step 2: Determine the function of each introduced tool regarding turning nuts.**
 *  The first tool (power screwdriver) is primarily for driving screws, although with adapters, it could potentially turn nuts, it's not its main function as presented.
 *  The second tool (socket wrench/ratchet/torque wrench) is specifically designed for turning nuts, bolts, and specific fittings like oil filters.
 *  The third tool (cutters/strippers) is for cutting or stripping wires, not turning nuts.
 *  The fourth tool (Knipex Pliers Wrench) is designed with parallel jaws to grip and turn nuts and bolts effectively without rounding them, functioning like an adjustable wrench.
 *  The fifth tool (combination pliers) can grip nuts but is not ideal as it can damage them; its primary functions are gripping, bending, and cutting. It's generally not considered a primary tool for turning nuts when proper wrenches are available.

*   **Step 3: Count the tools specifically used for turning nuts based on their design and common function.**
 *  The second tool (socket wrench/ratchet) is used for turning nuts/fittings.
 *  The fourth tool (Knipex Pliers Wrench) is used for turning nuts.
 *  The first, third, and fifth tools are not primarily designed or presented for turning nuts.

*   **Step 4: Summarize the findings.**
 Two of the tools introduced in the video are used for turning nuts: the second tool (socket wrench/ratchet) and the fourth tool (Knipex Pliers Wrench).

**Answer: 2 (the second one and the fourth one)**

**Gemini-2.0-Flash(1fps/1200f)**: let's analyze the video to determine how many of the tools shown are used for turning nuts.
The tools are shown on the table in the following order:
1. A pair of pliers with serrated jaws. 2. A pair of cutting pliers. 3. A pair of Knipex pliers wrench. 4. A DeWalt gyroscopic inline screwdriver. 5. A Tempest oil filter torque wrench.

Now, let's determine which of these tools can be used for turning nuts:
1.The first pair of pliers with serrated jaws can be used to grip and turn nuts, especially if they are damaged or rounded.
2.The cutting pliers are designed for cutting and are not suitable for turning nuts.
3.The Knipex pliers wrench is specifically designed to grip and turn nuts and bolts.
4.The DeWalt gyroscopic inline screwdriver is designed for driving screws and is not suitable for turning nuts.
5.The Tempest oil filter torque wrench is designed for tightening oil filters, which have a nut-like shape.

Therefore, three of the tools shown can be used for turning nuts: the first pair of pliers, the Knipex pliers wrench, and the Tempest oil filter torque wrench.

Answer: 3

(**Error Category**: Knowledge / Common-sense Error
Due to a knowledge error, the model mistakenly identified the first pair of pliers (Snap-on cutting pliers) as suitable for handling nuts.)

Figure 29: An annotation example of Human-R in the domain "Electric & Crafting" and its error analysis.

Figure 30: An annotation example of Human-R in the domain "Education" and its error analysis.

**Domain: Household**

**Annotation**

**Question**: What caused the shaking of camera in the video?

**CoT**: The video shows a man using a multimeter to test a battery that is part of a collection of mechanical components, which, based on the blue cap marking for windshield washer fluid, is identified as being in a car's engine compartment. Initially, the battery voltage was 12.92V, which later increased to 14.12V, indicating that the engine had started and was generating power; the process of starting the engine typically produces vibrations, and these vibrations likely transmitted to the camera during the measurement process.

**Answer**: The camera shook because the engine started and its vibrations were transmitted to the camera.

**Answers**

**o3**:
**Step-by-step reasoning**

1. Look at the frames: we can see only one person's hands working with two multimeters—first on a workbench and later on a car battery.
2. There is no other moving object (no running engine, no machinery that might vibrate).
3. The camera position clearly changes between the workbench scene and the car-battery scene, and slight shifts also occur while the person's hands reach toward the multimeter.
4. These small jolts match the moments when the person's hands are busy, implying the camera is not fixed on a tripod but is being held or repositioned by the same person (or someone assisting).
5. Thus the shaking is best explained by normal hand movement of the person holding the camera.

**Answer: The person holding or moving the camera caused it to shake.**

(**Error Category**: Visual Element Extraction Error
**Missing Referred Evidences** – The model missed the fine-grained referred evidence: "the abrupt camera shake after the engine starts", and instead misidentified an earlier, semantically similar motion as the evidence mentioned in the query.)

**GPT-4o**:
To determine what caused the shaking of the camera in the video, let's analyze the provided frames step by step:

1. **Frames 1–2**:
   The first two frames show a stable scene on a workbench with some batteries and multimeters. There's no noticeable shaking or movement, which suggests the camera was steady at this point.
2. **Frames 3–4**:
   The setting changes to a car engine bay with a multimeter placed on the car battery. The environment remains steady.
3. **Frames 5–8**:
   The scene continues to remain steady with a hand checking the multimeter. The camera appears stable.
4. **Frames 9–24**:
   The person adjusts and attaches the multimeter leads to the battery terminals. During this period, there may be subtle movements, but the camera does not exhibit noticeable shaking.
**Conclusion**:
Since there are no frames indicating shaking or instability in the video camera, the question might be based on a misunderstanding or incomplete data.

Therefore, the final answer is: "Answer: No shaking observed."

(**Error Category**: Visual Element Extraction Error
**Missing Referred Evidences** – missing the referred evidence: "the shaking of camera",)

**Gemini-2.0-Flash**:
The camera shaking is likely due to the person holding the camera while also using their hands to operate the multimeter and connect it to the car battery. This would cause slight movements and vibrations that are transferred to the camera.

Therefore, the final answer is: "Answer: The person holding the camera while operating the multimeter and connecting it to the car battery."

(**Error Category**: Perception Error
The model hallucinated, incorrectly identified that the operator was holding the camera while starting the engine.)

Figure 31: An annotation example of Human-R in the domain "Household" and its error analysis.

## H THE USE OF LARGE LANGUAGE MODELS (LLMs)

In accordance with the ICLR guidelines, we disclose the use of Large Language Models (LLMs) as general-purpose assistive tools in the preparation of this manuscript. Our use of LLMs was confined to two primary areas. First, during the initial literature review phase, we employed LLMs to assist in identifying and collecting relevant prior work and to generate concise summaries of several papers to expedite our understanding of the current research landscape. All referenced literature was subsequently read and critically analyzed by the authors. Second, in the manuscript writing process, LLMs served as an advanced writing aid for sentence polishing to improve clarity and flow, as well as for conducting grammar and spelling checks. We want to emphasize that the core intellectual contributions of this work, including the formulation of research questions, the development of our proposed methodology, the design of experiments, and the analysis and interpretation of results, were carried out exclusively by the human authors. The LLM was not used for generating key ideas, hypotheses, or conclusions, and therefore does not meet the criteria for authorship. The authors take full and final responsibility for all content presented in this paper.

