# OpenReview forum: "HumanPCR: Probing MLLM Capabilities in Diverse Human-Centric Scenes"
_ICLR.cc/2026/Conference — ICLR 2026 Poster_

### Official Review · Reviewer_9u8b · 2025-10-26

**Soundness:** 3
**Presentation:** 3
**Contribution:** 3
**Rating:** 6
**Confidence:** 3

**Summary:**

The paper introduces HumanPCR, a human-centric benchmark for evaluating multimodal large language models at three levels: perception, comprehension, and reasoning. It contains over 6,000 multiple-choice questions across 34 fine-grained human-related skills. These reasoning questions require the model to combine evidence from multiple moments in a video and notice important details that are not directly pointed out in the question. By testing 30+ state-of-the-art models, the paper shows that even the strongest systems still fall well short of human performance on fine-grained spatial understanding, temporal reasoning, and intent inference.

**Strengths:**

1. The paper is well written and organized.
2. The graph in the paper is clear and beautiful.
3. The core idea of this paper is interesting and special. It fills the gaps in prior benchmarks, which were either too narrow or too broad without focusing on human-centric understanding.
4. The dataset of this paper is carefully built with multi-stage human review and strict quality control. Each Human-R question must rely on multiple visual clues and include at least one piece of proactively gathered evidence, and only about 20% of questions are accepted.
5. Extensive experiments have been conducted to demonstrate the effectiveness. The paper evaluates more than 30 state-of-the-art models and shows that humans still outperform the best model by over 15%, while many open-source models score below 30% accuracy on the most challenging Human-R reasoning tasks.

**Weaknesses:**

1. Evaluation setup is not perfectly fair. The paper evaluates many models, but they are run with different inference settings, so the comparison is not strictly fair. For comprehension, models only get 32 sampled frames and must answer directly, while for reasoning some models can use the maximum number of frames they can handle and are given stronger prompts plus tricks like Best-of-N sampling.
2. Causal claims about failure modes are a bit strong. The paper concludes that proactive evidence extraction is the key bottleneck in complex video reasoning, based on manual analysis of 200 Human-R questions from top models; however, this is still correlational and not backed by controlled ablations.
3. The paper does not report the actual compute cost (runtime, GPU memory usage, or GPU hours) required to run the benchmark, which makes reproducibility and practical adoption harder.

**Questions:**

My questions have already been described in the Weaknesses section. I would like the authors to respond to these points in detail and clarify how they plan to address them.

---

> ### Author Response · Authors · 2025-11-24
> **Rebuttal by Authors(Part 1)**
>
> > **Response to Weakness1: Evaluation Fairness**
>
> Thank you for your thoughtful feedback. We would like to clarify our evaluation protocol, which was designed to ensure fairness relative to the distinct requirements of each task level:
>
> - **Distinct Task Requirements:** The Perception/Comprehension (Human-P/C) and Reasoning (Human-R) levels probe fundamentally different capabilities and are not directly comparable. Therefore, we aligned settings within each task level to match the task nature.
> - **Frame Numbers:** For Human-P/C, we standardized inputs to 32 frames because performance on these tasks saturates with relatively short video length. In contrast, Human-R requires dense visual integration. Forcing a low frame count here would unfairly penalize long-context models (e.g., LongVILA) whose core advantage is processing extensive visual data. Therefore, we evaluated each model across multiple frame sampling settings within its budget and **selected the specific configuration that yielded the best performance** for each individual model (as detailed in **Appendix C.1**). This ensures that every model is evaluated at its optimal capacity without artificial constraints.
> - **Prompts and “Tricks”:** To ensure a uniform comparison, we use the **same simple CoT prompt** for all models to obtain the main results (Table 3). Techniques like Best-of-N sampling are **not** used in the main evaluation results; they appear only in the ablation studies (Table 4) to analyze model sensitivity and upper bounds.
>
> > **Response to Weakness 2: Causal Claims About Failure Modes.**
>
> We appreciate the reviewer’s rigorous perspective on distinguishing correlation from causation. We acknowledge that without controlled intervention, claims regarding failure mechanisms can be speculative. In complex visual reasoning, the core difficulty lies in *selecting* the correct evidence—both content and perspective—from the raw video stream under conditioned probability. Therefore, to strictly verify that the failure to extract proactive evidence is the causal bottleneck, we conducted a new graded intervention study during the rebuttal that progressively lowers the difficulty of evidence selection:
>
> *   **Level 1 (Relation Awareness):** Prompting the model with potential relation types relevant to the scene (e.g., "check surrounding context" or "temporal antecedents").
> *   **Level 2 (Logic Awareness):** In addition to L1, indicating which specific *referred* evidence in the question typically holds logical connections to *proactive* evidence.
> *   **Level 3 (Proactive Guidance):** In addition to L2, providing **vague descriptions** of the specific proactive evidence to loosely guide the visual search.
>
> The results, presented in the table below, reveal a distinct pattern. The gains from L1 and L2 are marginal, indicating that generic attention guidance or logical hints are insufficient; the models remain unable to bridge the gap to the implicit cues. In contrast, L3 yields significant improvements.
>
> | Model                | Original Acc. |     Level 1 Acc.     |     Level 2 Acc.     |      Level 3 Acc.      |
> | :------------------- | :-----------: | :------------------: | :------------------: | :--------------------: |
> | **o4-mini**          |     53.39     | 52.26 (-1.13)        | 54.07 (+0.68)        | 63.35 (+9.96)          |
> | **GPT-4o**           |     41.40     | 41.62 (+0.22)        | 44.11 (+2.71)        | 52.35 (+10.95)          |
> | **Gemini-2.5-Flash** |     43.44     | 41.40 (-2.04)        | 45.93 (+2.49)        | 53.40 (+9.96)          |
> | **Qwen2.5-VL-72B**   |     34.39     | 34.61 (+0.22)        | 38.46 (+4.07)        | 47.74 (+13.35)         |
>
>
> This contrast is consistent with our view that **proactive evidence extraction** is a key practical bottleneck on Human-R: performance improves substantially only when we **directly support this step**, whereas purely relational or logical **hints for refered evidence** yield limited gains. Although this intervention does not isolate all possible factors, it provides more targeted support for this interpretation than our original correlational analysis. **We have incorporated these results into Section 4.3 and Table 6 and now frame them as supporting evidence about failure modes on Human-R, softening broader claims accordingly**.

---

> > ### Author Response · Authors · 2025-11-24
> > **Rebuttal by Authors(Part 2)**
> >
> > > **Response to Weakness3: Computational Cost**
> >
> > Thank you for pointing out the need for computational cost reporting to ensure reproducibility and practical adoption. We have measured the actual resource consumption for the full HumanPCR evaluation suite (including Human-P, Human-C, and Human-R).
> >
> > All experiments were conducted on a node with **8$\times$ NVIDIA A100 (80GB) GPUs** using the LMMs-Eval framework. Below, we report the specific runtime and GPU hours for representative models at the 7B, 38B, and 72B scales:
> >
> > | Representative Model | Size | Inference Configuration                  | Wall-Clock Time | Total GPU Hours(A100) |
> > | :------------------- | :--- | :--------------------------------------- | :-------------- | :-------------------- |
> > | **Qwen2.5-VL-7B**    | ~7B  | Data Parallel (**8** streams)            | ~0.8 hours      | ~6.4 hours            |
> > | **InternVL3-38B**    | ~38B | Model Parallel (**2** GPUs per instance) | ~12.5 hours     | ~25.0 hours           |
> > | **Qwen2.5-VL-72B**   | ~72B | Model Parallel (**3** GPUs per instance) | ~16.0 hours     | ~48.0 hours           |
> >
> > *Note on Optimization:* The reported times above utilize standard HuggingFace implementations. Practical deployment can be significantly accelerated using high-efficiency inference engines. While we report the baseline costs here for standardization, our released codebase will include support for these acceleration engines to facilitate faster evaluation for the community.
> >
> > We  include this detailed cost analysis in Appendix F.4 to support future research.

---

> ### Author Response · Authors · 2025-11-27
> **Follow-up on rebuttal and revised manuscript**
>
> Dear Reviewer 9u8b,
>
> I hope this message finds you well. We wanted to briefly check whether there are any remaining concerns regarding our rebuttal and revised manuscript. We would be happy to address any additional points you may have.
>
> Thank you again for your positive assessment and for your time and effort in reviewing our paper.
>
> Best regards,
> The authors

---

### Official Review · Reviewer_c88T · 2025-10-29

**Soundness:** 2
**Presentation:** 2
**Contribution:** 2
**Rating:** 4
**Confidence:** 3

**Summary:**

The HumanPCR dataset was constructed by first defining a three-tiered human-centric skill framework (Perception, Comprehension, Reasoning), and then sourcing video and image data from both public datasets (e.g., Ego4D, ActivityNet, MOMA) and recent web videos (e.g., YouTube clips from the last two years) spanning 11 real-world domains. For Human-P and Human-C, the team created over 6,000 multiple-choice questions using a mix of template generation, LLM drafting (e.g., GPT-4), and human review. For Human-R, they developed 442 open-ended video reasoning questions that require multiple pieces of visual evidence—at least one of which must be proactively retrieved, not directly cued—each paired with expert-authored Chain-of-Thought rationales. Rigorous quality control was applied, including blind filtering of text-answerable items, expert meta-review.

**Strengths:**

1) A reasoning paradigm that enforces multi-evidence integration and proactive evidence seeking, with rigorous filtering to avoid shortcutting.
2) Most datasets just ask questions where the answer is directly visible in one frame or clip. This benchmark forces models to search multiple places in the video to figure things out—like humans would.

**Weaknesses:**

1) The paper says annotations will be CC BY 4.0, but the DUA also says “non-commercial academic only.” CC BY allows commercial use—this must be reconciled (e.g., CC BY-NC for annotations, or remove NC language from DUA).
2) All answers are scored by a single LLM, even if correlations with humans are decent. To be rigorous, they should show confidence intervals, try multiple judges, and share prompts/seeds for transparency.
3) Only ~10% of Human-R questions were answered by actual humans. To make a strong comparison, they should show how many annotators were used, what instructions they had, and how reliable those answers are.
4) Many tables just give single accuracy numbers. They should report error bars, task-level variances, and statistical tests to show if differences are meaningful.

**Questions:**

1) How do you define “proactive” vs. “referred” evidence? Is this consistently labeled across annotators?
2) Do models improve because they see more, or because they choose better?

---

> ### Author Response · Authors · 2025-11-24
> **Rebuttal by Authors(Part 1)**
>
> > **Response to Weakness 1: Data License**
>
> Thank you for pointing out the inconsistency. We have revised the paper to **remove the statement** that annotations are released under CC BY 4.0. Instead, we now clearly state that access and use are governed by our Data Use Agreement (DUA). **The DUA provides CC BY-NC–like flexibility for non-commercial academic use**, while adding ethical safeguards prohibiting re-identification, surveillance, and biometric analysis. This reconciles the license language and aligns the release with responsible use.
>
> > **Response to Weakness 2: Concerns regarding Single LLM Judge, Confidence Intervals, and Transparency.**
>
> We must correct a factual oversight in this review: our submission **already** incorporates the multiple-judge evaluation and transparency measures requested.
>
> *   **Multiple Judges & Transparency (Already Present):** As explicitly analyzed in **Section 4.4** and **Figure 10**, we evaluated with **four distinct judges** (o3-mini, GPT-4o, Gemini-2.5-Flash, and o3). These judges demonstrated high consistency with human annotations (Pearson correlations $>0.64$) and stable relative rankings. Regarding transparency, we explicitly refer the reviewer to **Appendix C.2**, where **Figures 17–21** document all prompts referenced in the methodology. To ensure reproducibility, all judges used greedy decoding (`temperature=0`), effectively acting as a fixed seed.
> *   **Confidence Intervals (Added):** To directly address the reviewer’s concern, we re-ran all 8 models three times under the same o3-mini judge, and report the mean accuracy and **95% confidence intervals** (in the form mean ± δ) in below Table. The ranking are fully consistent with Table 3 in the paper, and the CI half-widths (δ) are mostly below 1 percentage point, indicating very limited variance across runs. This further supports that our single-LLM judge produces stable, reliable scores and that the relative ordering of models is robust to stochasticity in the evaluation process.
>
> | Rank | Judge (LLM) | Judged Model     | Avg Accuracy (%) | 95% CI (±) | Rank Consistency|
> | :--- | :---------- | :--------------- | -----------: | ---------: |  -------------: |
> | 1    | o3-mini     | o3               | 59.05        | ±2.03      |    True|
> | 2    | o3-mini     | gemini2.5pro     | 51.28        | ±0.86      |True|
> | 3    | o3-mini     | gpt4o            | 40.65        | ±0.65      |True|
> | 4    | o3-mini     | claude-3.7       | 40.42        | ±0.97      |True|
> | 5    | o3-mini     | gemini-2.0-flash | 38.73       | ±0.86      |True|
> | 6    | o3-mini     | internvl3-78b    | 36.82       | ±1.30      |True|
> | 7    | o3-mini     | qwenvl-2.5-72b   | 34.09        | ±0.32      |True|
> | 8    | o3-mini     | aria             | 28.45        | ±0.86      | True|
>
> > **Response to Weakness 3: Human-R Human Evaluation Details.**
>
> We appreciate the reviewer’s insightful comment and **would like to clarify that the requested details about Human-R are reported in Appendix C.4 and explain more details below**.
>
> For Human-R, we randomly sampled about 10% of the questions and asked **two independent evaluators** (graduate students who did not participate in dataset construction) to answer each sampled question; the **human accuracy reported in Table 3** is the **average** of these two annotators’ scores.
>
> Evaluators followed a controlled **text-only “open-book”** protocol. They were allowed to use a web search engine **only for static text resources** (dictionaries, encyclopedias, translation tools) to **translate non-native language phrases and clarify domain-specific terminology**, while access to video platforms, image-search services, and frame-based reverse image search was strictly prohibited, and browser histories were monitored to ensure compliance. This design not only addresses the practical needs of non-native evaluators, but also better reflects **realistic usage conditions**, where typical users naturally consult text resources for linguistic and conceptual clarification while still relying on the given video for **visual reasoning**.

---

> ### Author Response · Authors · 2025-11-24
> **Rebuttal by Authors(Part 2)**
>
> > **Response to Weakness 4: Missing Statistical Tests**
>
> We thank the reviewer for suggestions of quantifying uncertainty beyond single-point accuracy. To address this, for representative models, we have run each experiment three times to assess run-to-run variability. These results in the table below indicate that the scores are highly stable across runs, making the single-point estimates reliable.
>
> - **More detailed results of every level and dimension, are reported in Appendix F.3 (Table 11).**
>
> |                               | Human-P        | Human-C        | Human-R        |
> | ----------------------------- | -------------- | -------------- | -------------- |
> | ***Qwen2.5-VL-72B***          |                |                |                |
> | Avg $\pm$ (95% CI half-width) | 57.90$\pm$0.28 | 54.75$\pm$0.61 | 34.31$\pm$1.41 |
> | Variance                      | 0.01           | 0.06           | 0.32           |
> | ***GPT-4o***                  |                |                |                |
> | Avg $\pm$ (95% CI half-width) | 48.62$\pm$2.62 | 50.45$\pm$2.41 | 41.02$\pm$0.86 |
> | Variance                      | 1.11           | 0.94           | 0.12           |
>
>
> > **Response to Question 1: Definition and Consistency of "Proactive" vs. "Referred" Evidence**
>
> We appreciate this insightful question. The distinction between "proactive" and "referred" evidence is the cornerstone of Human-R. To ensure this is objectively quantifiable and reproducible, we established a rigorous definition and a standardized annotation protocol involving **LLM assistance, weighted scoring, and multi-stage verification**.
>
> **1. Clear Definitions based on Information Gap**
> We define the two types based on the semantic coverage of the question text over the visual scene:
> *   **Referred Visual Evidence:** Visual information that is **explicitly pointed to** in the question (e.g., *Question: "What is the man in the red shirt holding?"* → The "man in the red shirt" is referred; the model performs grounding).
> *   **Proactive Visual Evidence:** Visual information that is **necessary for reasoning but absent from the question text**. The model must deduce the *need* for this information based on task logic (e.g., *Question: "Why did the car stop?"* → The model must proactively search and select visual condidates like a "traffic light," which is not mentioned in the text).
>
> **2. Quantification and "Degree of Proactivity" (Consistency Mechanism)**
> We acknowledge that the boundary can be subtle (e.g., indirect hints). Instead of a simple binary decision, we implemented a **fine-grained weighted protocol**:
>
> *   **Step 1 (LLM-Assisted Decomposition):** First, an LLM extracts a candidate list of necessary visual evidence (atomic facts) based on the human-authored Chain-of-Thought reasoning.
> *   **Step 2 (Human Refinement & Scoring):** Annotators verify and correct this list (applying deduplication rules for semantic redundancy). Crucially, for each piece of evidence, they assign a **"proactive degree" scalar** based on how much the question reveals about it (e.g., 0 for fully mentioned, higher scores for unmentioned).
> *   **Step 3 (Weighted Aggregation):** The final "count" of proactive evidence is not a count of raw items passed a threshold, but a **sum of these weighted scores**. This method scientifically handles the "gray area" of partial hints, ensuring the metric reflects the true reasoning demand.
>
> **3. Quality Control for Alignment**
> Since no absolute "gold standard" exists for semantic interpretation, we relied on strict consensus mechanisms:
> *   **Dual-Review & Cross-Validation:** Each annotation was reviewed by at least **two independent reviewers**. They cross-checked the evidence list and scores; any discrepancies were flagged.
> *   **Meta-Review Resolution:** Senior Main Authors (Meta-Reviewers) acted as arbitrators to resolve these flagged conflicts, ensuring the final "proactive" classification aligns with our definitions rather than individual annotator bias.

---

> ### Author Response · Authors · 2025-11-24
> **Rebuttal by Authors(Part 3)**
>
> > **Response to Question 2: "see more" vs "choose better"**
>
> This is a profound question that touches the very core of our analysis in **Section 4.3**. Our experiments clearly indicate that while "seeing" is a prerequisite, **"choosing better" (via proactive reasoning) is the primary bottleneck and determinant of performance** on Human-R.
>
> We address this dialectical relationship through below findings:
>
>  1.  **"Seeing More" has diminishing returns:** As shown in **Figure 6**, simply scaling up frame counts (inputting more visual data) yields negligible improvements for complex reasoning tasks. For questions requiring 6+ pieces of evidence, adding frames can even be detrimental due to distraction, proving that raw information intake is insufficient without the reasoning to filter it.
> 2.  **"Choosing Better" outweighs raw perception:** As noted in **Table 4** and **Figure 8**, reasoning-enhanced models like **o3**  significantly outperform standard models (e.g., Gemini-2.0-Flash) even when o3 processes **fewer frames** (e.g., 64 vs. 128). This demonstrates that the capacity to proactively deduce *what* to look for (reasoning-driven selection) is more critical than simply having *more* frames available.
> 3.  **Passive vs. Proactive Choice:** Furthermore, **Figure 7** shows that "choosing" enhanced methods (like retrieval-based RAG or token selection) only improves a little on Human-R. This confirms that the "choice" cannot be a simple query-match but must be a **reasoning-driven proactive search** for implicit cues.
> 4.  **Verification via Prompt Intervention (New Experiment):**
>     To definitively test "seeing" capability from "choosing" capability, we have conducted additional intervention experiments during the rebuttal. **We have integrated the new results and analysis into Section 4.3 and Table 6**. We tested three levels of guidance:
>   *   **Level 1 (Relation Awareness):** Prompting the model with potential relation types relevant to the scene (e.g., "check surrounding context" or "temporal antecedents").
>   *   **Level 2 (Logic Awareness):** In addition to L1, indicating which specific *referred* evidence in the question typically holds logical connections to *proactive* evidence.
>   *   **Level 3 (Proactive Guidance):** In addition to L2, providing **vague descriptions** of the specific proactive evidence to loosely guide the visual search.
>
> | Model                | Original Acc. |     Level 1 Acc.     |     Level 2 Acc.     |      Level 3 Acc.      |
> | :------------------- | :-----------: | :------------------: | :------------------: | :--------------------: |
> | **o4-mini**          |     53.39     | 52.26 (-1.13)        | 54.07 (+0.68)        | 63.35 (+9.96)          |
> | **GPT-4o**           |     41.40     | 41.62 (+0.22)        | 44.11 (+2.71)        | 52.35 (+10.95)          |
> | **Gemini-2.5-Flash** |     43.44     | 41.40 (-2.04)        | 45.93 (+2.49)        | 53.40 (+9.96)          |
> | **Qwen2.5-VL-72B**   |     34.39     | 34.61 (+0.22)        | 38.46 (+4.07)        | 47.74 (+13.35)         |             |
>
> As shown in above table,  **L1 and L2 hints yielded only marginal gains**, whereas **L3 brought significant improvements**. This pattern suggests that, once the search is loosely anchored to the right region, current models can exploit additional proactive visual evidence; the main limitation is not only low-level perception, but the ability to **locate, prioritize, and integrate** the right cues. L1/L2, which only provide abstract relational or logical hints in text, leave this visual search process largely unchanged, while L3 partially short-circuits it by vaguely indicating what the critical evidence is like. These support our view that the core bottleneck on Human-R is not “seeing more”, but running a visual-centric reasoning policy that can **choose** better evidence for problem solving.

---

> > ### Comment · Reviewer_c88T · 2025-11-24
> > **Concerns Addressed**
> >
> > The reviewer appreciates the authors' efforts and have raise scores accordingly.

---

> > > ### Author Response · Authors · 2025-11-25
> > > **Appreciation for the Follow-Up**
> > >
> > > Thank you very much for your encouraging follow-up and supportive assessment. We truly appreciate your thoughtful engagement with our rebuttal. We remain fully open to further discussion and to addressing any additional concerns or potential improvements that could strengthen the paper.

---

### Official Review · Reviewer_4oKa · 2025-10-31

**Soundness:** 3
**Presentation:** 2
**Contribution:** 3
**Rating:** 4
**Confidence:** 3

**Summary:**

This paper presents HumanPCR, a hierarchical benchmark for human-centric visual understanding in MLLMs, consisting of Human-P/C (over 6k multiple-choice items across 34 tasks and 9 dimensions) and Human-R (442 open-ended video reasoning questions spanning 11 domains). A central design of Human-R is to require multi-evidence integration and at least one proactive (non-referred) visual evidence per question, with expert-annotated CoT rationales and rigorous filtering to avoid shortcutting via query cues. Evaluating 30+ SOTA models, the authors show that current MLLMs substantially underperform humans, with pronounced weaknesses in fine-grained spatial/temporal understanding and mind modeling, and that mere frame/context scaling yields diminishing or negative returns when evidence requirements grow. In contrast, reasoning-oriented models and test-time Best-of-N sampling provide consistent gains. The benchmark includes quality control, prompts, model configs, LLM-judge validation against human judgments, and ethical data terms.

**Strengths:**

The benchmark’s most distinctive contribution is Human-R’s explicit requirement for multi-evidence reasoning with at least one proactive, non-referred cue—an evaluation target largely missing from existing video QA benchmarks. This forces models beyond query-matching shortcuts and surfaces a realistic capability gap in long, complex, human-centric videos. The taxonomy across Human-P/C delivers both breadth and diagnostic depth; macro-averaging at task/dimension level makes failure modes actionable (notably posture, contact, procedure). The annotation and verification design is strong: expert-authored open-ended questions, evidence-grounded CoT rationales, blind filtering to remove textual shortcuts, and meta-review to ensure non-redundancy and proactive evidence. The empirical study is comprehensive: 30+ models, analyses of frame scaling, multiple context-extraction strategies, test-time scaling (Best-of-N vs self-refine), and a human-grounded error taxonomy highlighting proactive evidence omissions. LLM-judge reliability is quantified against 4k human ratings with high correlations and no detectable self-bias, supporting practical reproducibility. Overall, the work addresses an important and under-evaluated capability—human-centric multi-evidence reasoning—of clear significance to assistive systems, egocentric AI, and HRI.

**Weaknesses:**

While Human-P/C are valuable and thorough, their task space overlaps with prior perception/comprehension benchmarks; the strongest novelty lies in Human-R. The paper would benefit from ablations demonstrating added diagnostic value of Human-P/C design (beyond coverage), or clearer evidence that specific sub-tasks reduce confounds seen in earlier datasets. Human-R’s reliance on an LLM judge, albeit validated, still risks metric drift as judges evolve; a small, fully human-adjudicated gold subset would strengthen longitudinal stability and cross-paper comparability. The proactive-evidence constraint is compelling but mainly supported by qualitative/error analyses; controlled ablations—e.g., converting proactive evidence into referred cues, or masking question hints—could quantify its causal contribution to difficulty and discriminative power. Finally, given aggregation from public datasets and web videos, a more explicit analysis of potential pretraining overlap with popular MLLM corpora would help dispel contamination concerns and clarify generalization.

**Questions:**

Can the authors provide controlled variants of Human-R items to quantify the marginal difficulty induced by proactive evidence, for example by (i) turning proactive evidence into referred cues, and (ii) adding minimal hints? Reporting Δ accuracy across models would causally validate the role of proactivity and identify which architectures benefit most. To disentangle perception/extraction vs reasoning/knowledge, have the authors run oracle-evidence experiments that supply the required clip timestamps or bounding boxes? Such upper bounds would localize the dominant bottleneck and guide method development. For judge robustness, would the authors consider releasing a small human-adjudicated “gold set” (balanced across domains and reasoning types) with reference CoTs to calibrate future LLM judges and track metric stability over time? Also, did judge agreement vary systematically by reasoning type (e.g., planning vs causal) or domain (e.g., education vs sports)? Given that adding frames can hurt when evidence count is high, have the authors tried structured evidence-gathering policies—iterative timeline search, subgoal decomposition, or tool-augmented retrieval—to mitigate attention dilution? Even preliminary results would clarify whether the bottleneck is context management or search policy. Finally, since several tasks touch on emotions and social relations, have the authors conducted bias or robustness audits across demographics and cultural contexts? A brief analysis would strengthen ethical readiness for real-world, human-facing applications.

---

> ### Author Response · Authors · 2025-11-24
> **Rebuttal by Authors(Part 1)**
>
> > **Response to Weakness 1: Overlap/Novelty of Human-P/C**
>
> - **1. On the novelty of Human-P/C vs prior benchmarks.**  We respectfully disagree with the claim that Human-P/C “mainly overlaps” with existing perception/comprehension benchmarks and that the strongest novelty lies only in Human-R. Human-P/C is explicitly designed to *fill gaps* in human-centric ability coverage rather than to re-label prior tasks: it spans 9 dimensions and 34 tasks over both images and videos, while prior human-centric benchmarks typically cover at most 1–4 dimensions and often a single modality. In addition, each task is sourced from *diverse* datasets rather than a single benchmark, specifically to reduce single-dataset biases and avoid simply reproducing the idiosyncrasies of any one source.
> - **2. Fine-grained human-centric skills, not just more tasks.** Beyond scale, the *structure* of Human-P/C differs from prior work. Our taxonomy explicitly isolates fine-grained human-centric skills that are either absent or collapsed in existing benchmarks, such as gaze and body-orientation estimation, detailed contact types (human–object, human–human, self-contact), and long-term procedure and social-relation understanding. Existing benchmarks usually report only coarse scores like “action” or “relation”, which cannot disentangle preferences across closely related but distinct abilities (e.g., self-contact vs human–object contact, sequential vs goal-driven procedures).
>
> - **3. Evidence that Human-P/C adds diagnostic value.** Human-P/C is not just “more coverage”: it enables diagnostics that coarse task spaces cannot provide. In our new-added results (Sec. 4.4, Figure 9), (i) model rankings change substantially across neighboring subtasks within the same high-level dimension, and (ii) the effectiveness of CoT depends jointly on task complexity and model strength, with CoT helping mainly on more fine-grained, visually demanding tasks and sometimes hurting on simpler ones. This heterogeneity directly supports the *non-redundancy* of our taxonomy and shows that collapsing these subtasks into a single “action” or “relation” score would hide systematic failure modes.
> - **4. Human-P/C and Human-R are complementary.** Finally, Human-R is *complementary* to, rather than the sole source of novelty in HumanPCR. Human-P/C provides a unified, human-centric ability space with fine-grained probes across perception and comprehension; Human-R then builds on this space to stress-test models under multi-evidence, proactive reasoning. Together, they offer a level of coverage and diagnostic power that, to the best of our knowledge, is not available in existing human-centric or video-reasoning benchmarks.

---

> > ### Author Response · Authors · 2025-11-24
> > **Rebuttal by Authors(Part 2)**
> >
> > > **Response to Weakness 2, Question 3 and Question 4: Gold Set, Judge Robustness and Agreement Analysis**
> >
> >   We appreciate these insightful suggestions to ensure the longevity and reliability of our evaluation metrics.
> >
> >   **1. Release of Human-Adjudicated "Gold Set"**
> >   We commit to releasing a **human-adjudicated "gold set"** to facilitate future calibration and metric stability tracking. This dataset will include `question_id`, full Q&A content, diverse model responses, and the corresponding human judgments.
> >
> >   **2. Judge Agreement by Type and Domain**
> >   We conducted a breakdown analysis of the Pearson correlation between LLM judges and human annotators across different reasoning types and domains. The results are summarized below:
> >
> > **Table 1: Judge Agreement by Reasoning Type (Accuracy Pearson / Win–loss Pearson)**
> >
> > | Reasoning Type   | Accuracy Pearson (o3) | Win–loss Pearson (o3) | Accuracy Pearson (o3-mini) | Win–loss Pearson (o3-mini) |
> > | :--------------- | :-------------------: | :-------------------: | :------------------------: | :------------------------: |
> > | Planning         |         0.444         |         1.000         |           0.444            |           1.000            |
> > | Causal Reasoning |         0.568         |         0.848         |           0.593            |           0.835            |
> > | Prediction       |         0.758         |         0.915         |           0.739            |           0.922            |
> > | Counter-Factual  |         0.578         |         0.667         |           0.659            |           0.769            |
> > | Assessment       |         0.753         |         0.950         |           0.749            |           0.867            |
> >
> > **Table 2: Judge Agreement by Domain (Accuracy Pearson / Win–loss Pearson)**
> >
> > | Domain                       | Accuracy Pearson (o3) | Win–loss Pearson (o3) | Accuracy Pearson (o3-mini) | Win–loss Pearson (o3-mini) |
> > | :--------------------------- | :-------------------: | :-------------------: | :------------------------: | :------------------------: |
> > | Electric & Crafting          |         0.560         |         0.880         |           0.591            |           0.809            |
> > | Education                    |         0.500         |         0.682         |           0.547            |           0.677            |
> > | Sports                       |         0.785         |         0.937         |           0.721            |           0.894            |
> > | Household                    |         0.746         |         0.947         |           0.721            |           0.952            |
> > | Indoor Leisure & Self Care   |         0.562         |         0.652         |           0.635            |           0.742            |
> > | Outdoor Leisure & Recreation |         0.707         |         0.891         |           0.701            |           0.884            |
> > | Outdoor Adventure            |         0.756         |         0.929         |           0.845            |           0.935            |
> > | Transport                    |         0.750         |         0.633         |           0.858            |           0.750            |
> > | Daily Spending               |         0.649         |         0.612         |           0.668            |           0.764            |
> > | Performance & Exhibition     |         0.513         |         0.765         |           0.513            |           0.743            |
> > | Others                       |         0.870         |         1.000         |           0.935            |           1.000            |
> >
> >
> >   As indicated in the tables, judge agreement does vary systematically. We observe a performance drop in:
> >     1.  **Complex Reasoning Types:** Tasks like *Planning*, which require forward-looking logic and multi-step deduction, show lower correlation (\~0.56) compared to more retrospective tasks.
> >     2.  **Fine-Grained Domains:** Domains requiring specialized visual expertise or handling minute details, such as *Electric & Crafting*, also present higher difficulty for the text-based judge (\~0.60).
> >
> > While these specific subsets challenge the LLM judge—reflecting the inherent difficulty of evaluating high-level reasoning and domain-specific verification—the judge maintains a strong positive correlation in the majority of categories. The overall alignment remains robust, suggesting that while the judge is slightly less sensitive in the most extreme test cases, it remains a reliable proxy for aggregate human preference in diverse human-centric scenarios.

---

> > > ### Author Response · Authors · 2025-11-24
> > > **Rebuttal by Authors(Part 3)**
> > >
> > > > **Response to Weakness 3, Question 1 and Question 2: Controlled Ablations on Proactive vs. Referred evidence."**
> > >
> > > We thank the reviewer for the suggestion and first clarify our conceptual stance. In long, cluttered human-centric videos, locating the right clips is not a detachable pre-processing step but an integral part of the reasoning process: the model must decide eivdence involved upon perception. Even for seemingly “referred” cues (e.g., “when she puts the phone down”), resolving the correct visual support in minutes-long video already requires temporal disambiguation, coreference, and scene understanding rather than simple query matching. Naively “turning proactive evidence into referred cues” or supplying oracle snippets would thus alter the task into text-guided retrieval or text-only reasoning over human-selected evidence, precisely the shortcut behavior that Human-R is designed to rule out.
> > >
> > > Our original analyses (Fig. 7 and Fig. 8) already indicate that (i) merely scaling frames or using query-guided context extraction provides limited gains, and (ii) errors are dominated by visual evidence extraction, especially missed proactive cues, rather than misperception of attended frames. To provide more direct causal support—without collapsing proactive search into a separate “module”—we ran a search-guidance intervention on a subset of Human-R during rebuttal and added related analysis to Section 4.3 and Table 6. Instead of rewriting questions or feeding oracle timestamps, we progressively strengthened how much the question guides evidence search while keeping visual reasoning intact:
> > >
> > > - Level 1 (Relation awareness): we hint at **relevant relation types** (e.g., “consider earlier causes” or “check surrounding context”).
> > > - Level 2 (Logic awareness): L1 + we indicate **which referred evidence** in the question typically connects to proactive cues.
> > > - Level 3 (Proactive guidance): L2 + a **vague natural-language description of the proactive evidence** (without giving the answer).
> > >
> > > | Model                | Original Acc. | Level 1 Acc.  | Level 2 Acc.  |  Level 3 Acc.  |
> > > | :------------------- | :-----------: | :-----------: | :-----------: | :------------: |
> > > | **o4-mini**          |     53.39     | 52.26 (-1.13) | 54.07 (+0.68) | 63.35 (+9.96)  |
> > > | **GPT-4o**           |     41.40     | 41.62 (+0.22) | 44.11 (+2.71) | 52.35 (+10.95) |
> > > | **Gemini-2.5-Flash** |     43.44     | 41.40 (-2.04) | 45.93 (+2.49) | 53.40 (+9.96)  |
> > > | **Qwen2.5-VL-72B**   |     34.39     | 34.61 (+0.22) | 38.46 (+4.07) | 47.74 (+13.35) |
> > >
> > > Across all models, L1/L2—loosely analogous to making the question more “referred” and relation-aware—yield only marginal or negative gains, whereas Level 3 consistently delivers an absolute gain of about 10–13% once the proactive cue is *loosely described* in language. This pattern indicates that models can correctly interpret and integrate the proactive frames once gently pointed to them (“seeing” and local visual reasoning are largely intact), but they fail to autonomously bridge from the question to those implicit cues. Combined with the error breakdown in Fig. 8, this intervention causally supports our claim that the primary bottleneck is **reasoning-driven evidence selection and integration**, rather than abstract language-only reasoning.
> > >
> > > Regarding “oracle evidence” via timestamps or bounding boxes, our current annotations are based on language-level CoTs rather than dense temporal/bbox labels, so a clean large-scale oracle study is beyond the scope of this release. We view timestamp/bbox-based oracle benchmarks as complementary future work, while Human-R deliberately targets the practically critical ability to *find and choose* the right evidence in long videos, not just to reason once that evidence has been handed to the model.
> > >
> > > > **Response to Weakness 4: Pretraining Overlap and Data Contamination**
> > >
> > > We employed a rigorous multi-layered safeguard strategy. **1) Data Provenance:** For Human-P/C, we strictly sampled from the validation or test splits of established peer-reviewed datasets to maximize isolation from training sets. For Human-R, we collected web videos published within the last two years, explicitly filtering out IDs present in major video datasets (e.g., Panda-70M, MOMA) to minimize overlap. **2) Novel Annotation:** Crucially, regardless of raw media provenance, the evaluative core—our QA pairs—is newly created and distinct from pretraining captions. We implemented "blind filtering" to remove questions answerable by text-only knowledge, ensuring that models cannot rely on memorized facts or priors. While the opacity of modern training corpora makes exhaustive audits infeasible—a common challenge in the community—our combination of clean-split usage, recent source filtering, and novel, reasoning-heavy question formulation effectively mitigates leakage risks.

---

> ### Author Response · Authors · 2025-11-24
> **Rebuttal by Authors(Part 4)**
>
> > **Response to Question 5: Structured Evidence-Gathering Policies:**
>
> We would like to clarify that **we have indeed conducted extensive experiments on such structured evidence-gathering policies**, as explicitly presented in **Figure 7**, **Table 9**, and discussed in **Section 4.3 (Page 8)**.
>
> Specifically, to investigate whether advanced configurations could mitigate the attention dilution you mentioned, we evaluated a wide range of methods that align with the "iterative search" and "tool-augmented retrieval" categories:
>
> *   **Memory & Retrieval (Tool-augmented/Iterative):** We benchmarked **Video-RAG**, **$T^*$**, **Dispider**, and **IXC2.5-OmniLive** (see Table 9 for details). These methods explicitly utilize retrieval mechanisms to locate relevant visual cues.
> *   **Structured Frame/Token Selection:** We tested **FRAG**, **AKS**, **FlexSelect**, and **LongVU**, which employ structured policies to select key frames or tokens.
> *   **Internal Reasoning/Decomposition:** We evaluated **Video-R1**, **Video-RFT**, and **o3**, which utilize internal "thinking" processes (Chain-of-Thought/Subgoal decomposition) to structure the reasoning path.
>
> As shown in Fig. 7 and detailed in Appendix C.1, structured evidence-gathering methods deliver clear relative gains on Video-MME but much smaller improvements on Human-R—especially single-heuristic variants—and even test-time reasoning strategies (e.g., Best-of-N and specialized reasoning models) still fail to overcome the core bottleneck of integrating proactive multi-evidence in complex human-centric scenes.
>
> This empirical result directly answers your question: we *have* tried these policies, and their failure underscores the unique challenge of Human-R—it requires proactive, holistic reasoning that cannot be solved by simple heuristic matching or existing retrieval-augmented pipelines.
>
> > **Response to Question 6: Bias and Ethical Safeguards:**
>
> We ensured ethical readiness through rigorous source selection and active curation. **1) Vetted Sources:** For sensitive dimensions like *Emotion* and *Social Relation*, we rely exclusively on widely accepted, peer-reviewed datasets that have already cleared ethical screening. Moreover, we have attempted to aggregate multiple data sources within each task to minimize the bias inherent in any single dataset. As these sources have satisfied publication standards, conducting a retrospective bias audit on established benchmarks is neither standard practice nor the objective of this work. **2) Rigorous Review:** For Human-R, we implemented a multi-stage manual review. During data collection, any meaningless, harmful, or illegal content was proactively identified and removed by human screeners. Furthermore, annotators and reviewers were mandated to reject questions involving stereotypes, discriminatory language, or time-sensitive private information. **3) Usage Control:** We enforce a strict Data Use Agreement (DUA) explicitly prohibiting biometric identification, surveillance, or non-research misuse. These measures ensure HumanPCR assesses human-centric capabilities without violating privacy standards.

---

> ### Author Response · Authors · 2025-11-27
> **Follow-up on rebuttal and revised manuscript**
>
> Dear Reviewer 4oKa,
>
> I hope this message finds you well. We wanted to follow up to see whether the clarifications in our rebuttal and the additional experiments in the revised draft have adequately addressed your concerns. If you are satisfied, we would kindly ask you to consider updating your evaluation to reflect the new results and discussion. We remain fully committed to addressing any remaining points you may have.
>
> Thank you very much for your time and effort in reviewing our paper.
>
> Best regards,
> The authors

---

### Official Review · Reviewer_dhYK · 2025-10-31

**Soundness:** 3
**Presentation:** 3
**Contribution:** 3
**Rating:** 8
**Confidence:** 4

**Summary:**

The paper proposes a fairly reasonable pipeline to curate complex VLM evaluation benchmarks across different difficulty levels (Figure 3).
The paper takes extra care to ensure that the quality and difficulty of the dataset is high. For instance, the questions solvable by blind LLM are filtered. In addition, the reasoning questions that one evidence, or lack of reasoning were not kept.  The paper tests a wide range of models in the open and closed world across several model scales. The fine-grained analysis reveals that only reasoning model o4-mini improves consistently with more visual context while the other models fail to do so.

**Strengths:**

1. The paper proposes a fairly reasonable pipeline to curate complex VLM evaluation benchmarks across different difficulty levels (Figure 3).

2. The paper takes extra care to ensure that the quality and difficulty of the dataset is high. For instance, the questions solvable by blind LLM are filtered. In addition, the reasoning questions that one evidence, or lack of reasoning were not kept.

3. The paper tests a wide range of models in the open and closed world across several model scales. The fine-grained analysis reveals that only reasoning model o4-mini improves consistently with more visual context while the other models fail to do so.

**Weaknesses:**

1. The process of human participation is not very clear at different stages of the work. (a) who are domain experts and for what domains? (b) Where are these domain experts sourced from and how many of them were there? (c) Did the same folks also solve the task for human evaluation number in Table 3? (d) Is human eval taken from the same kind of domain experts that made the dataset or are they average human workers on some platform?

2. The writing-style and presentation could be improved. For instance, (a) it is pretty hard to read some figures like 7 and 8. (b) the spacing seems to be altered at many places – hardly any gap between paragraphs on page 3 and 8.

**Questions:**

Mentioned above

---

> ### Author Response · Authors · 2025-11-24
> **Rebuttal by Authors**
>
> > **Response to Weakness 1: Clarification on Human Participation**
>
> We thank the reviewer for pointing out the need for greater clarity regarding human participation. We provide the details below.
>
> **(a) & (b) Domain Experts in Data Construction.**
>
> As detailed in Appendix B.3, Human-R covers 11 diverse domains. Our recruitment strategy prioritized domain relevance over a rigid “one-size-fits-all” standard:
>
> - **For specialized domains** (e.g., *Sports, Electric & Crafting, Transport*), we recruited annotators with relevant practitioner experience or educational backgrounds (e.g., fitness enthusiasts for sports, engineering-related training for crafting) to ensure that the questions and reasoning chains reflect plausible domain logic.
> - **For general domains** (e.g., *Household, Daily Spending*), we engaged high-quality annotators capable of precise common-sense reasoning.
> - **Sourcing.** The dataset was constructed by a dedicated team of annotators recruited specifically for this project, who were assigned to domains aligned with their background and passed qualification trials following our annotation guidelines.
>
> **(c) & (d) Human Evaluation Participants & Protocol.**
> We appreciate the reviewer’s insightful comment and **would like to clarify that the requested details about Human-R are reported in Appendix C.4 and explain more details below**.
>
> The human evaluation in Table 3 was conducted by a **separate group of evaluators** (primarily graduate students) who did not participate in dataset construction or annotation, in order to avoid any data leakage.
>
> For Human-R, we adopted a controlled **“open-book”** protocol to approximate realistic usage, where human users (including non-native English evaluators) can consult text resources while MLLMs already possess extensive internalized knowledge from pre-training. Evaluators were allowed to use a web search engine **only for static, text-based resources** (e.g., dictionaries, encyclopedias, translation tools), **primarily to translate non-native language phrases and to clarify domain-specific terminology** (such as *“carburetor”* or *“belay device”*). Access to video platforms, image-search services, and frame-based reverse image search was prohibited.
>
> In practice, participants only occasionally used search and mostly for translation or clarification of rare terms. Thus, search functioned as a lightweight aid for **language understanding** rather than as a source of additional multimodal evidence. The reported human accuracy therefore still primarily reflects **visual-centric reasoning** over the provided video, with only minimal support for factual and linguistic disambiguation. If we removed this limited text-only access, we would expect human accuracy to decrease slightly on terminology-heavy questions while remaining clearly above current MLLMs, so the qualitative conclusion that humans substantially outperform models on HumanPCR would remain unchanged.
>
> >  **Response to Weakness 2: writing-style and presentation**
>
> Thank you for the helpful suggestions regarding writing style and presentation.
>
> - Figures 7–8: We have redesigned these figures to improve readability by tightening the layout, increasing the font sizes of axis labels, tick marks, and legends so that the comparisons are clearer.
> - Page 3: We have increased the spacing between paragraphs in the RELATED WORK section and added clearer separation around the Taxonomy introduction.
> - Page 8: We have enlarged the spacing above and below the “Takeaway” in Section 4.3 to avoid overcrowding.
>
> We believe these changes address the readability and spacing concerns and improve the overall presentation.

---

> ### Author Response · Authors · 2025-11-27
> **Follow-up on rebuttal and revised manuscript**
>
> Dear Reviewer dhYK,
>
> I hope this message finds you well. We wanted to briefly check whether there are any remaining concerns regarding our rebuttal and revised manuscript. We would be happy to address any additional points you may have.
>
> Thank you again for your positive assessment and for your time and effort in reviewing our paper.
>
> Best regards,
> The authors

---

### Author Response · Authors · 2025-11-24
**Summary of Paper Revisions**

We thank the reviewers for their thoughtful reviews and constructive feedback; we appreciate the time and effort they invested, which has helped us improve the clarity and rigor of our submission. Summary of our Paper Revisions is as follows:


1. **Quantitative proactive-evidence intervention study** (Reviewer 4oKa, 9u8b, c88T):

   Added graded intervention of evidence extraction hints experiments of Human-Rto test model reasoning mode on proactive evidence of Human-R in **Sec. 4.3** and **Table 6**.

2. **CoT effects and fine-grained task differences on Human-P/C** (Reviewer 4oKa):

   Added analysis of CoT effects and changes in accuracy rankings across neighboring subtasks to highlight fine-grained task differences on Human-P/C, in **Sec. 4.4** and **Fig. 9**.

3. **Report standard deviation, and confidence intervals on HumanPCR** (Reviewers 4oKa, c88T):

   Added 3-run standard deviation and 95% CIs for representive models in **App. F.3 (Table 11)**.

4. **Computational cost reporting** (Reviewer 9u8b):

   Reported wall-clock time and total GPU hours for representative 7B/38B/72B models on the full suite in **App. F.4**.

5. **Human annotation & evaluation protocol details** (Reviewers c88T, dhYK):

   Clarified domain-expert recruitment for Human-R construction, and the text-only “open-book” protocol for human baselines in **App. B.4** and **App. C.4**.

6. **Licensing and DUA clarification** (Reviewer c88T):

   Removed the CC BY 4.0 statement and clarified that data access and use are governed by our Data Use Agreement (DUA) in **Appendix E (Figure 25)**.

7. **Presentation and figure readability** (Reviewer dhYK):

   Improved layout and font sizes for **Figs. 7–8**, increased spacing in **Related Work (page. 3)**, and around the **Takeaway in Sec. 4.3 (page. 8)** to enhance readability.

---

### Author Response · Authors · 2025-12-02
**Clarification of Core Contributions of HumanPCR**

For context, we briefly restate the core contributions of HumanPCR as clarified in our revised manuscript. HumanPCR addresses two critical gaps of MLLM benchmarks: the lack of **fine-grained granularity** in human-centric tasks and the absence of **proactive multi-evidence video reasoning** in complex scenarios. We bridge these gaps through three splits of HumanPCR:

**1. Human-P and Human-C: Redefining Resolution in Human-Centric Understanding**
Current benchmarks often aggregate performance into coarse "action" or "relation" scores, obscuring model deficiencies. Human-P and Human-C solve this via:

* **Unprecedented Granularity:** We establish the first unified taxonomy covering **34 fine-grained tasks across 9 dimensions** (Fig. 11), enabling diagnosis at a resolution prior benchmarks cannot achieve.
* **Task-Driven Realism:** By curating data from **diverse sources (ego/exo views, daily/professional scenes)** (Fig. 3) explicitly matched to tasks, we mitigate single-source bias and yield realistic coverage of human-centric scenarios..
* **Deep Diagnostic Value:** Our analysis reveals that neighboring subtasks exhibit **heterogeneous CoT gains** (Fig. 9a) and **ranking changes**(Fig. 9b), proving that Human-P and Human-C successfully **disentangle subtle model capabilities** that are lost in traditional aggregate metrics.

**2. Human-R: Pioneering the "Proactive Visual Reasoning" Paradigm**
Moving beyond reasoning with refered evidence tested in existing works, Human-R opens a new direction for **open-ended, multi-evidence reasoning**:

* **Paradigm Shift to Open-Endedness:** As the **first open-ended video reasoning benchmark** (Tab. 1), Human-R mandates **proactive visual reasoning**  independent of multiple-choice cues, thereby eliminating the "option-leakage" bias prevalent in existing datasets.
* **Rigorous Complexity:** With a ~20% acceptance rate for expert-curated questions (App. B.3.2), we ensure genuine **visual complexity**(Fig. 2,5,6), **evidence-reasoning necessity**(Tab. 3,4,6), and **minimal text/single-frame bias**(Tab. 5).
* **Identifying the Next Frontier:** We demonstrate that **scaling context length or heuristic hints/context extraction alone does not solve Human-R** (Fig. 6,7, Tab. 6). This identifies **proactive evidence extraction** as a distinct bottleneck, offering the community a clear roadmap for future MLLM improvements.

---

### Meta-Review · Area_Chair_28kf · 2026-01-08

**Summary:**

1. Causal support for the “proactive reasoning” bottleneck:
The reviewers felt that the original manuscript’s conclusion—that failures are driven mainly by proactive evidence extraction—was stated too strongly given that much of the supporting evidence was correlational rather than controlled.

2. Judge robustness, statistical uncertainty, and longitudinal stability:
The reviewers raised concerns about reliance on LLM-based judging, potential metric drift as judges evolve, and the lack of stronger uncertainty reporting in the main results.

3. Fairness of evaluation settings and compute transparency:
The reviewer questioned whether comparisons are strictly fair across models and task levels due to differences in inference settings.

4. Novelty and incremental value of Human-P/C vs prior benchmarks
The reviewer felt their novelty relative to existing perception/comprehension benchmarks needed clearer evidence.

**Reviewer Concerns:**

Addressed Concerns:

1. Causal support for the “proactive evidence” bottleneck:
The rebuttal adds a graded intervention study. The pattern—minimal/negative gains for L1/L2 but consistent +10–13% gains for L3—provides substantially stronger evidence than the original correlational/error analysis that difficulty is driven by finding/selecting implicit evidence.

2. LLM-judge robustness, uncertainty reporting, and long-term stability:
The authors clarify/extend multi-judge evaluation and provide additional judge–human agreement analyses, including breakdowns by reasoning type and domain.

3. Evaluation fairness and comparability across models/settings:
The authors clarify that main results use a consistent baseline CoT prompt, and that techniques like Best-of-N appear in ablations rather than the primary leaderboard.

**Reviewer Scores:**

None

---

### Decision · Program_Chairs · 2026-01-26

Accept (Poster)